# Perinatal murine cytomegalovirus infection reshapes the transcriptional profile and functionality of NK cells

Carmen Rožmanić[1], Berislav Lisnić[1], Marina Pribanić Matešić[1], Andrea Mihalić[1], Lea Hiršl[1], Eugene Park[2], Ana Lesac Brizić[1], Daniela Indenbirken[3], Ina Viduka[4], Marina Šantić[4], Barbara Adler[5], Wayne M. Yokoyama[2], Astrid Krmpotić[6], Vanda Juranić Lisnić[1], Stipan Jonjić [1]✉ & Ilija Brizić [1]✉

Infections in early life can elicit substantially different immune responses and pathogenesis than infections in adulthood. Here, we investigate the consequences of murine cytomegalovirus infection in newborn mice on NK cells. We show that infection severely compromised NK cell maturation and functionality in newborns. This effect was not due to compromised virus control. Inflammatory responses to infection dysregulated the expression of major transcription factors governing NK cell fate, such as Eomes, resulting in impaired NK cell function. Most prominently, NK cells from perinatally infected mice have a diminished ability to produce IFN-γ due to the downregulation of long non-coding RNA *Ifng-as1* expression. Moreover, the bone marrow's capacity to efficiently generate new NK cells is reduced, explaining the prolonged negative effects of perinatal infection on NK cells. This study demonstrates that viral infections in early life can profoundly impact NK cell biology, including long-lasting impairment in NK cell functionality.

Natural killer (NK) cells are members of the innate lymphoid cell (ILC) family and play an important role in the control of tumors and intracellular pathogens[1]. NK cells express a repertoire of germline-encoded activating and inhibitory receptors, determining how an NK cell will respond to a target cell. NK cells display considerable functional and phenotypic overlap with ILC1 cells, and both cell types in mice are defined as lineage⁻NK1.1⁺NKp46⁺ lymphocytes. Integrin CD49b and the transcription factor Eomes expressed by NK cells, and integrin CD49a expressed by ILC1 cells are commonly used to distinguish these two cell types. Both cell types show a potent ability to produce IFN-γ, but ILC1 cells have limited or no cytotoxic ability. Unlike ILC1 cells, whose development is primarily independent of the bone marrow, NK cells develop in the bone marrow, from where they enter the bloodstream[2].

Even though these cell types are considered separate lineages, NK cells have been reported to acquire ILC1-like properties under certain pathological conditions, a process mediated by TGF-β or IL-12[3,4].

After egress from the bone marrow, mouse NK cells undergo the final stages of maturation in peripheral organs. NK cell functional maturation steps are usually defined by the expression of CD27 and CD11b[5]. Immature CD27⁺CD11b⁻ cells give rise to double-positive cells, which differentiate into mature CD27⁻CD11b⁺ cells. Subsequently, mature NK cells acquire KLRG1 expression and are considered terminally mature. During their maturation, NK cells increase their cytotoxic capacity while decreasing cytokine production and expansion ability[5]. Mechanisms regulating NK cell maturation, particularly during inflammation, are not fully understood[6].

[1]Center for Proteomics, Faculty of Medicine, University of Rijeka, Rijeka, Croatia. [2]Division of Rheumatology, Department of Medicine, Washington University School of Medicine, St. Louis, MO, USA. [3]Heinrich Pette Institute, Leibniz Institute for Experimental Virology, Hamburg, Germany. [4]Department of Microbiology and Parasitology, University of Rijeka, Faculty of Medicine, Rijeka, Croatia. [5]Max von Pettenkofer Institute & Gene Center, Virology, Faculty of Medicine, LMU Munich, Munich, Germany. [6]Department of Histology and Embryology, Faculty of Medicine, University of Rijeka, Rijeka, Croatia. ✉e-mail: stipan.jonjic@medri.uniri.hr; ilija.brizic@medri.uniri.hr

Genetic and environmental factors shape the NK cell repertoire[7]. While in some cases the adaptation of NK cells to extrinsic factors is considered favorable for the host, in other cases the changes in NK cell repertoire can lead to a state of exhaustion[8]. Cytomegalovirus (CMV) infection is a well-established example of immune system remodeling by the pathogen, especially in the NK cell compartment. The most striking change in the NK cell repertoire induced by human CMV (HCMV) is the accumulation of terminally differentiated NKG2C[+] NK cells with adaptive features[9,10]. It is not known to what extent NKG2C[+] NK cells control infection. In adult C57BL/6 mice, NK cell activating receptor Ly49H can directly recognize the mouse CMV (MCMV) protein m157, thereby providing a signal for the direct elimination of infected cells[11,12]. NK cells expressing the activating receptor Ly49H expand upon MCMV infection and form a pool of adaptive NK cells with an enhanced ability to control reinfection[13].

Age can significantly affect the course and outcome of infections. Differential age-dependent immune responses to infections are particularly pronounced in intrauterine and early postnatal life, including the reduced functional capacity of human neonatal NK cells[14,15]. Again, HCMV is not an exception: while infection in healthy adults is usually asymptomatic, a congenital infection can cause an array of morbidities and even death[16]. Increased susceptibility to congenital HCMV infection is partially due to the inability of the immature immune system to cope with infection and subsequent inflammation. The properties and behavior of NK cells in early life, especially in the context of infections, are not well understood. While adult NK cells are constantly generated in the bone marrow and enter the bloodstream, in mice NK cells appear in the first week after birth[17]. Furthermore, immaturity of NK cells, associated with a reduced Ly49H expression, results in a higher burden of MCMV upon infection of young mice[18]. How the infection in the context of an immature and developing immune system affects the NK cell compartment remains so far unknown.

In this study, we demonstrate that MCMV infection of newborn mice significantly affects the phenotype of NK cells, dysregulates the expression of major NK cell transcription factors, and results in impaired NK cell function. Furthermore, we show that inflammatory cytokines IL-12 and IL-18, induced by infection, are able to mediate functional and transcriptional changes in NK cells. At the same time, the bone marrow's capacity to generate new NK cells is reduced, explaining the prolonged adverse effects of perinatal infection on NK cells.

## Results

### Low expression of Ly49 receptors is a hallmark of NK cells during ontogeny and renders NK cells incapable of controlling MCMV infection

NK cells are considered to be immature in newborn mice. However, cellular and molecular characterization of NK cell immaturity in newborn mice is lacking. To address this issue, we have performed transcriptomic analysis of sorted NK cells from mice at post-natal days (PND) 7, 14 and 21, and from adult mice (60 days old). A dynamic pattern of gene expression changes in NK cells during this period was observed (Fig. 1a). The major characteristic of NK cells in the early postnatal period is increased regulation of genes involved in the cell cycle (clusters (CL) 2, 3 and 4), in line with the extensive process of proliferation of NK cells required to fill organ niches during ontogeny[19]. These findings were confirmed by analysis of Ki67 expression on postnatal days 14 and 60, which indicated drastically higher levels of Ki67 in NK cells of young, 14 days old mice as compared to adult mice (Fig. 1b). In addition to genes involved in cell proliferation, differential regulation was observed for CL1, which includes genes involved in the regulation of immune function. Interestingly, and in line with observations made by others[18,20], our data indicated that expression of genes coding for Ly49 receptors was lower in NK

cells of young mice (Fig. 1c). Accordingly, protein expression of different Ly49 receptors, including Ly49H, Ly49A and Ly49I, was significantly lower on NK cells of young mice as shown by flow cytometry (Fig. 1d). NK cells mediate direct Ly49H receptor-dependent control of MCMV infection in adult C57BL/6 mice[12,21,22]. To understand the functional implications of NK cell immaturity during ontogeny, we infected C57BL/6 mice either on PND 1, PND 7 or PND 14 with MCMV, and analyzed virus titers in the spleen, liver and lungs of control and NK cell-depleted groups of mice 4 days post infection (p.i.) (Fig. 1e, Fig. S1a-b). NK cell depletion did not impact virus titer if mice were infected on PND 1 and PND 7. However, NK cells provided significant control when infection was performed on PND 14 (Fig. 1e, Fig. S1b), in accordance with the increase in Ly49H expression (Fig. 1d). Altogether, we show that during the postnatal period, mouse NK cells exhibit a sequential decrease in their cell cycle-related gene signature while acquiring functional capacity, as evidenced by an increase in the expression of Ly49 receptors.

### NK cells are activated following perinatal MCMV infection

Since MCMV replicates for weeks after infection of newborn mice, we have next investigated the impact of NK cells on the viral replication kinetics. Following infection on PND 1, the virus is cleared from the spleen by day 21 p.i. in both the control and NK cell-depleted group of mice (Fig. 2a). In accordance with results shown in the previous section, depletion of NK cells did not result in a significant increase of the virus load in the spleen on day 7 and 12 p.i. Similarly, NK cells had no impact on virus titer in the brain, liver and lungs on day 7 and 12 p.i., while, as in the spleen, the virus was cleared from these organs by day 21 (Fig. S2), altogether arguing against a major role of NK cells in the control of MCMV following infection of newborn mice. The numbers of NK1.1[+] cells were significantly reduced on day 12 post-infection (p.i.) in spleens of infected mice. However, they reached the same levels as in control mice by 21 days p.i., when replicating virus is below the detection limit (Fig. 2b). An increased frequency of Ly49H[+] NK cells in infected mice was observed on day 21 p.i., suggesting an infection-induced expansion of Ly49H[+] NK cells (Fig. 2c), as shown in adult mice[13]. Even though NK cells did not control the virus in the early days p.i., they still produced IFN-γ and upregulated the activation marker CD69 in response to virus replication in the spleen (Fig. 2d, e). Altogether, these results demonstrate that NK cells are activated during MCMV infection of newborn mice despite their inability to control virus replication.

### Perinatal MCMV infection drives terminal maturation of NK cells

Productive MCMV infection of newborn mice is resolved in the spleen by day 21 p.i. (Fig. 2a). To determine if the NK cell compartment was restored to the same state as in control mice, we determined the maturation status of splenic NK cells by analyzing the expression of CD27 and CD11b[5]. The majority of splenic NK cells from infected mice were of the terminally differentiated phenotype (CD27[−]CD11b[+]), in contrast to control animals in which this population was small (Fig. 3a). Furthermore, NK cells in infected mice showed a high expression of KLRG1, another well-established marker of terminal NK cell maturation (Fig. 3b)[2,23]. Strikingly, the frequency of terminally differentiated NK cells was higher in infected animals at all analyzed time points, even 40 days post-infection (Fig. 3c, d) and despite virus clearance. A similar shift in NK cell maturation status was observed in other tissues, such as the lungs, liver, and blood (Fig. 3e, f). Since changes in NK cell maturation are usually associated with broader changes in phenotype, we have analyzed an array of NK cell markers on day 21 p.i. (Fig. S3). The expression of many NK cell receptors and molecules important for their function was lower in NK cells in MCMV infected mice, including activation receptors DNAM-1, NKp46 and Ly49D, the chemokine receptor CXCR3, the integrin DX5, the adhesion molecule CD62L, the

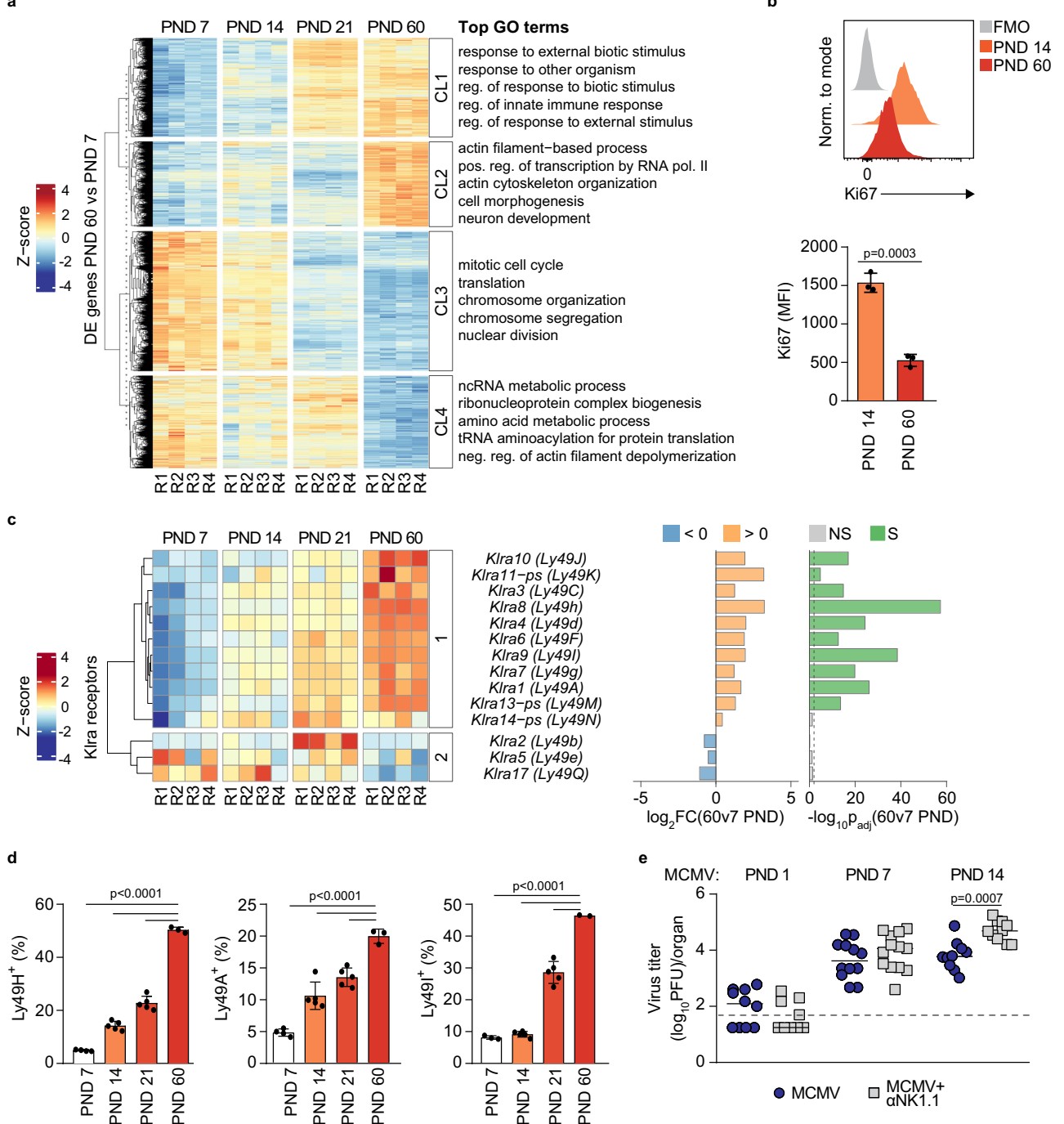

**Fig. 1 | Immaturity of murine NK cells during the postnatal period. a** Splenic NK cells were sorted on post-natal days (PND) 7, 14, 21 and 60. Following RNASeq differential expression analysis, k-means clustering was performed on the set of differentially expressed genes ($p_{adj} < 0.01$) between PND 60 and PND 7 using a predetermined value of $k = 4$. Genes in identified clusters (CL1 to CL4) were then subjected to enrichment analysis, and the five most significant gene ontology (GO) terms were listed next to each cluster ($n = 4$ mice per time point). **b** At indicated time points, NK cell expression of Ki67 was determined using flow cytometry ($n = 3$ mice per time point, Two-tailed Student's $t$ test). **c** RNASeq heatmap showing the expression of *Klra* family genes encoding the surface NK cell receptors on PND 7, 14, 21 and 60. K-means clustering was performed using a predetermined value of $k = 2$. Graphs illustrating $\log_2 FC$ and negative $\log_{10} p_{adj}$ values for the differences in expression levels on PND 60 versus PND 7 for each gene are shown to the right of the heatmap (NS = non-significant, S = Significant). **d** At indicated time points, NK cell expression of Ly49H, Ly49A and Ly49I was determined using flow cytometry ($n = 4$ PND 7, $n = 5$ PND 14 and PND 21, $n = 3$ PND 60, One-way ANOVA test). **e** Mice were infected at indicated time points, and the proportion of mice in each group was depleted of NK cells (αNK1.1). Viral doses used for infection were: 200 PFU for PND 1, 30 000 PFU for PND 7, and 50 000 PFU for PND 14. Virus titers were determined in the spleen 4 days post-infection ($n = 10$ PND 1 and PND 14, $n = 12$ PND 7 MCMV, $n = 13$ PND 7 MCMV + αNK1.1; pooled data of $N = 2$, Mann–Whitney two-tailed test). Statistically significant differences are indicated ($P$ values). Source data are provided as a Source Data file.

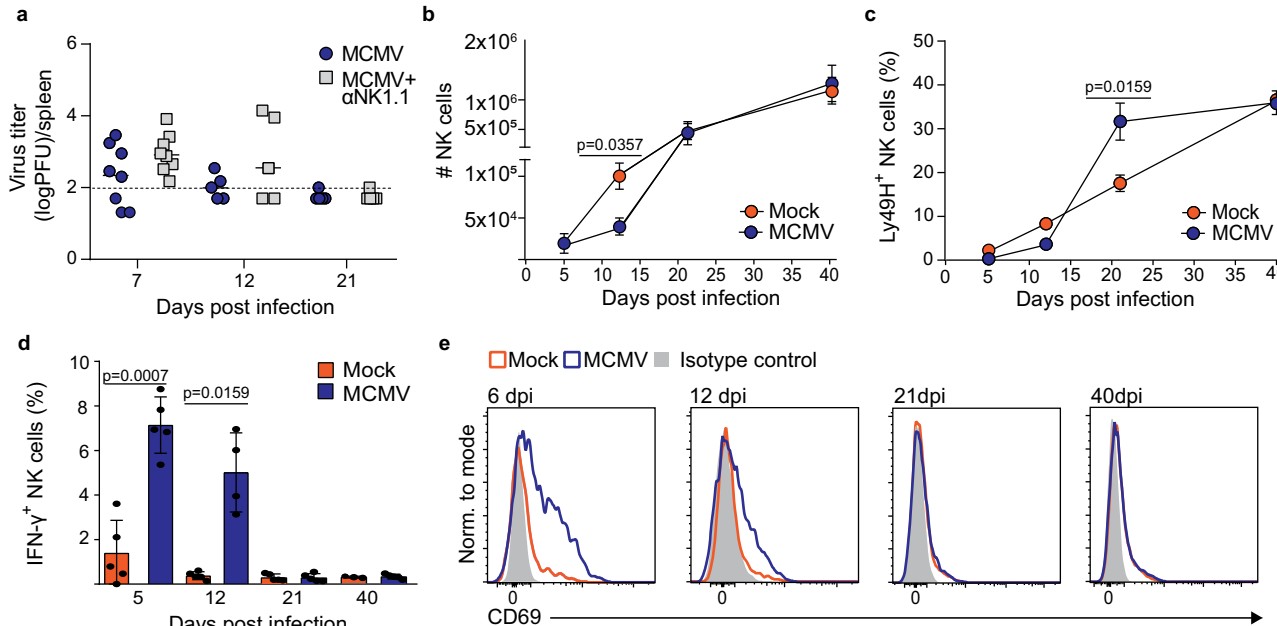

**Fig. 2 | NK cell response to MCMV infection in newborn mice.** Newborn C57BL/6 mice were intraperitoneally infected with 200 PFU of MCMV or mock-infected 24–36 h after birth. **a** The indicated groups of mice were depleted of NK cells (αNK1.1). At designated time points, virus titers were determined in the spleen (*n* = 8 (d.p.i. 7), *n* = 5 (d.p.i. 12 and d.p.i. 21.)). Dashed line represents detection limit. At indicated time points, the numbers of NK cells (**b**), the frequency of Ly49H⁺ NK cells (**c**), the frequency of IFN-γ producing NK cells (**d**) (*n* = 5 d.p.i. 5, d.p.i. 12, d.p.i. 21 and d.p.i. 40 MCMV; *n* = 3 d.p.i. 40 Mock, Mann–Whitney two-tailed test), and the expression of CD69 by NK cells (**e**) were analyzed in the spleen. Mean values ± SD are shown. Statistically significant differences are indicated (*P* values). Source data are provided as a Source Data file.

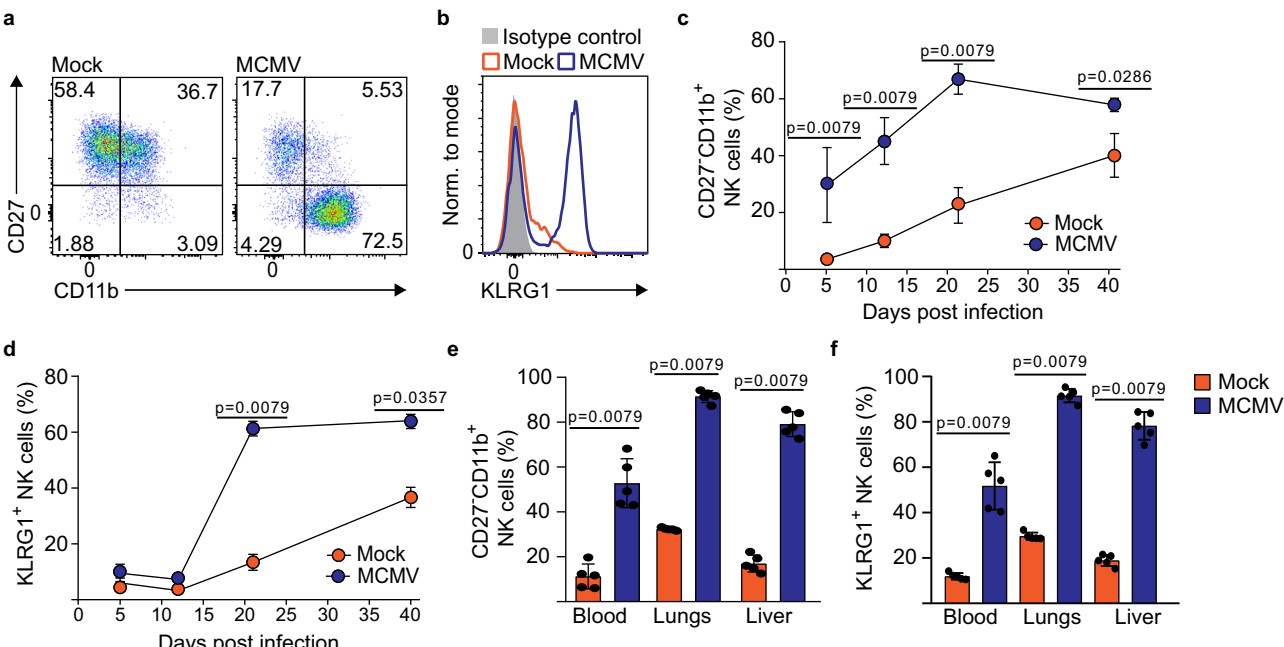

**Fig. 3 | Terminal maturation of NK cells following MCMV infection of newborn mice.** Newborn C57BL/6 mice were intraperitoneally infected with 200 PFU of MCMV or mock-infected 24–36 h after birth. On day 21 (**a**, **b**), or at indicated time points after infection (**c**, **d**), splenic NK cells were analyzed using flow cytometry. Expression of CD27, CD11b (**a**, **c**), and KLRG1 (**b**, **d**) on splenic NK cells is shown. **e, f** The frequency of terminally differentiated CD27⁻CD11b⁺ and KLRG1 expressing NK cells on day 21 p.i. in blood, lungs and liver. Representative dot plots (**a**), histograms (**b**) or mean values ± SD (**c**–**f**) are shown (*n* = 5 mice per group, Mann–Whitney two-tailed test). Statistically significant differences are indicated (*P* values). Source data are provided as a Source Data file.

interleukin-2 receptor subunit beta CD122, and the apoptosis regulator Bcl-2. Collectively, these data indicate that MCMV infection of newborn mice drives drastic changes in NK cell maturation which are associated with dysregulation of a substantial number of key NK cell receptors important for their function.

## Terminal maturation of NK cells in perinatally infected mice is characterized by downregulation of the T-box transcription factor Eomes

T-box transcription factors are major transcription factors that mediate NK cell development and maturation[2]. For this reason, we

investigated the expression of Eomes and T-Bet in NK cells from perinatally infected mice at day 21 p.i. The results of our study point out that Eomes protein level was downregulated in NK cells (Fig. 4). This downregulation was drastic, with a significant proportion of NK1.1⁺ cells from MCMV-infected mice not expressing Eomes at all and the rest displaying significantly reduced Eomes levels. At the same time, levels of T-bet, a transcription factor highly related to Eomes, were unaffected. Differences in Eomes expression were primarily

associated with the population of terminally differentiated NK cells (KLRG1⁺, CD27⁻CD11b⁺) (Fig. 4b, c). In contrast, less mature CD27⁺CD11b⁻ NK cells in MCMV-infected mice downregulated Eomes expression in a minor subset of cells while the levels of Eomes in those that retained the expression were identical to that in NK cells from uninfected mice. Similarly to infection on postnatal day 1, infection of mice on postnatal day 7 also resulted in terminal differentiation and downregulation of Eomes in terminally differentiated NK cells (Fig. S4).

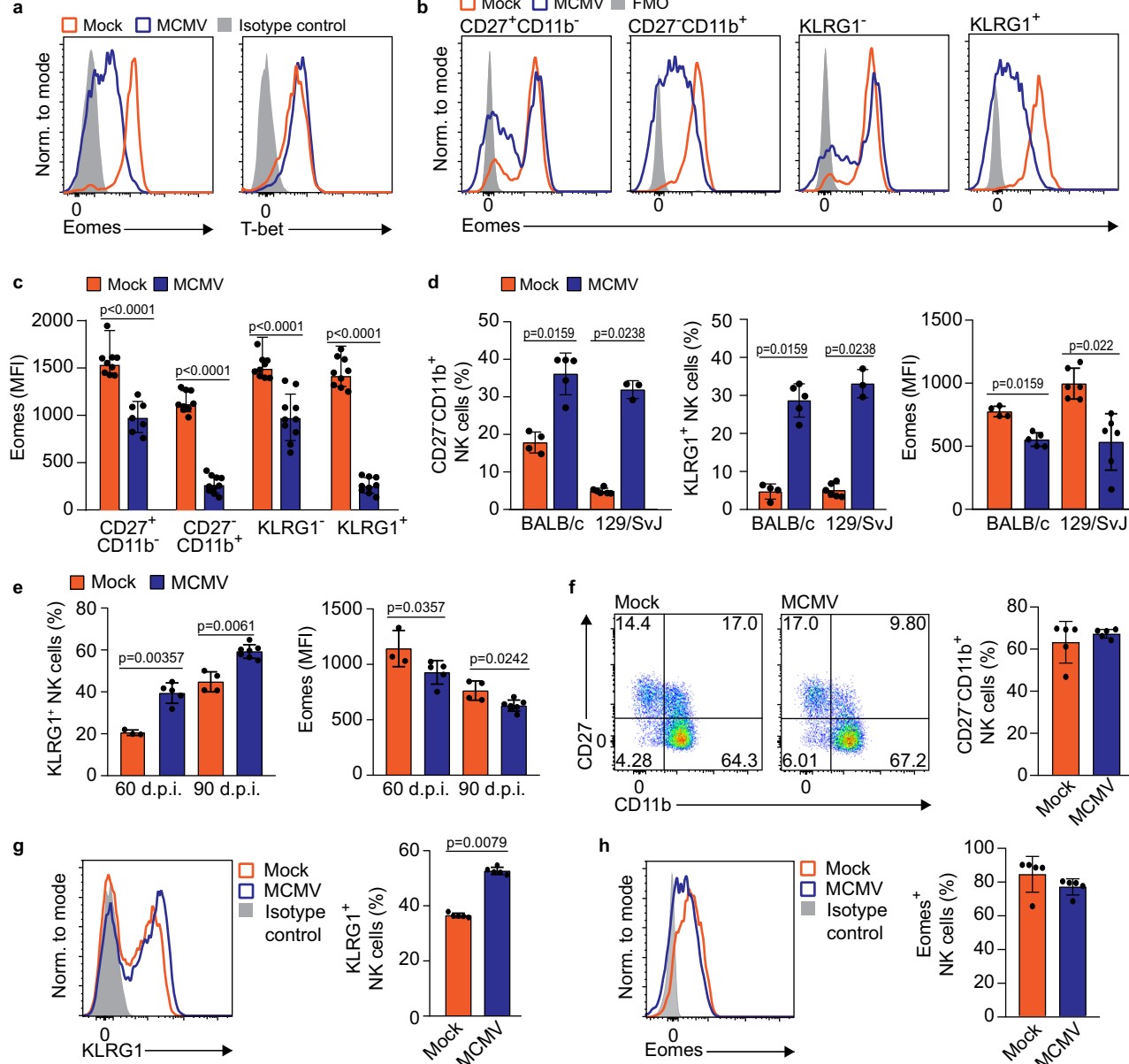

**Fig. 4 | Terminal maturation of NK cells is associated with the downregulation of Eomes. a–c** Newborn C57BL/6 mice were intraperitoneally infected with 200 PFU of MCMV or mock-infected 24−36 h after birth. **a** On day 21 after infection, expression of Eomes and T-bet was determined by flow cytometry. **b** Eomes expression was analyzed on CD27⁺CD11b⁻, CD27⁻CD11b⁺, KLRG1⁺ and KLRG1⁻ NK cells from MCMV and mock-infected mice, representative histograms are shown. **c** Quantification of Eomes expressing CD27⁺CD11b⁻, CD27⁻CD11b⁺, KLRG1⁺ and KLRG1⁻ NK cells (*n* = 9 mice per group, Mann−Whitney two-tailed test). Mean values ± SD are shown. **d** Newborn BALB/c and 129/SvJ mice were intraperitoneally infected with 200 PFU of MCMV or mock-infected 24−36 h after birth. On day 21 after infection, mice were euthanized and the frequency of terminally differentiated, KLRG1 and Eomes expressing splenic NK cells was determined (*n* = 4 Mock

BALB/C, *n* = 5 MCMV BALB/C, *n* = 6 Mock 129/SvJ, *n* = 3 MCMV 129/SvJ, Mann−Whitney two-tailed test). Mean values ± SD are shown. **e** Newborn C57BL/6 mice were intraperitoneally infected with 200 PFU of MCMV or mock-infected 24−36 h after birth. On days 60 and 90 after infection, mice were euthanized and the expression of KLRG1 and Eomes on NK cells was analyzed (*n* = 3 Mock 60 d.p.i., *n* = 4 MCMV 60 d.p.i., *n* = 4 Mock 90 d.p.i., *n* = 6 MCMV 90 d.p.i., Mann−Whitney two-tailed test). Mean values ± SD are shown. **f–h** Adult C56BL/6 mice were infected with 2 × 10⁵ PFU of WT MCMV. On day 21 p.i., the expression and frequency of CD27 and CD11b (**f**), KLRG1 (**g**) and Eomes (**h**) by NK cells were analyzed by flow cytometry (*n* = 5 Mock, *n* = 5 MCMV, Mann−Whitney two-tailed test). Representative histograms and mean values ± SD are shown. Statistically significant differences are indicated (*P* values). Source data are provided as a Source Data file.

To test whether NK cells were similarly affected in other strains of mice, we infected 129/SvJ and BALB/c mice with MCMV and also observed Eomes downregulation in NK cells accompanied by terminal maturation (Fig. 4d). To verify how long these changes are maintained, we analyzed NK cells 60 and 90 days post-infection and still observed differential expression of KLRG1 and Eomes (Fig. 4e). These data suggest that perinatal MCMV infection of newborn mice results in the emergence of terminally differentiated NK cells expressing low levels of the transcription factor Eomes.

To determine whether this profound downregulation of Eomes in NK cells during perinatal MCMV infection also occurs upon infection of adult mice, we analyzed NK cell maturation, KLRG1 and Eomes expression at day 21 following the infection of adult C57BL/6 mice (Fig. 4f–h). In adult mice, a significant proportion of NK cells are CD27−CD11b+[5]. However, in sharp contrast to newborn mice, we did not observe a robust change in NK cell maturation (Fig. 4f, g), nor significant downregulation of Eomes in MCMV-infected adult mice (Fig. 4h), which supports the finding that downregulation of Eomes is specific for MCMV infection of newborn mice.

## NK cell downregulation of Eomes following perinatal infection is independent of virus load but requires active virus replication

The NK cell receptor Ly49H mediates effective MCMV control in adult C57BL/6 mice. However, Ly49H is not significantly expressed in newborn mice, resulting in poor virus control by NK cells (Fig. 2a and Fig. S2). Therefore, the control of infection is impaired in newborn mice compared to adults and we hypothesized that the impaired virus control in newborn mice is the underlying cause of the observed profound changes in NK cells. To test this hypothesis, we infected

newborn mice with several doses of MCMV ranging from 50 to 800 PFU, since $LD_{50}$ for neonatal mice is ~1000 PFU of MCMV[24]. There were no significant differences in Eomes expression in NK cells between mice infected with different doses of MCMV (Fig. 5a). Furthermore, we employed highly attenuated MCMV strain, m74stop MCMV, which lacks the major entry complex gH/gL/gO, but can establish low-level infection due to use of alternative entry complex gH/gL/MCK-2[24,25]. Even an infection with m74stop MCMV induced a significant downregulation of Eomes, indicating that a low level of viral replication is sufficient to induce its downregulation in NK cells (Fig. 5b). Furthermore, we used recombinant MCMV which expresses MULT-1, a ligand for activating NK cell receptor NKG2D, and which is strongly attenuated in newborn mice in an NKG2D-dependent manner[26]. Similar to infection with m74stop MCMV, infection with MULT-1MCMV resulted in Eomes downregulation in NK cells (Fig. 5c). However, the levels of Eomes remained unchanged following injection of low or high dose of UV-inactivated MCMV into newborn mice, indicating that viral replication is required for Eomes downregulation (Fig. 5d). Similarly, PolyIC injection into newborn mice did not affect NK cells (Fig. S5a). These data raised a possibility that antiviral therapy could prevent downregulation of Eomes in NK cells. To this end, we treated infected newborn mice with ganciclovir and immune sera, which reduce viral replication (Figs. 5e, f, Fig. S5b). None of the treatments significantly prevented the downregulation of Eomes in NK cells. However, these treatments reduce virus levels but do not completely eliminate the virus allowing it to establish a productive infection in newborn mice. To test if the vaccination approach could prevent NK cell Eomes downregulation, we immunized female mice with MCMV and then infected their offspring (newborn mice) with

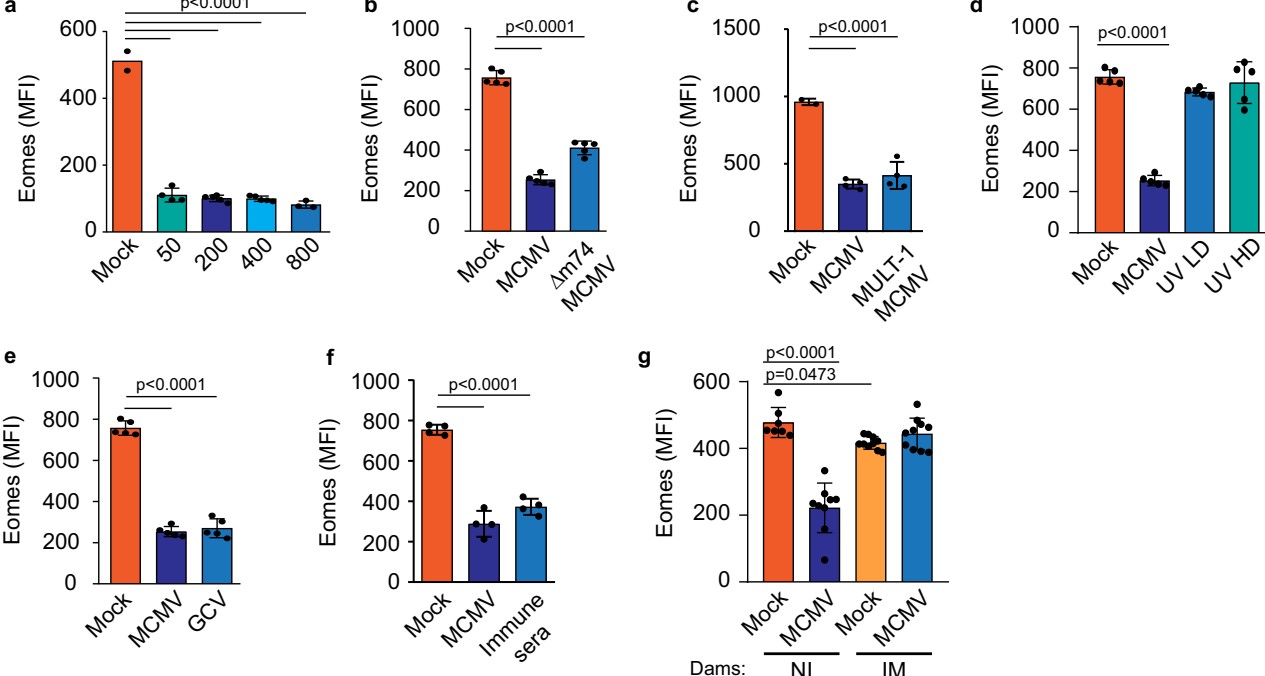

**Fig. 5 | Impact of virus load on NK cell Eomes downregulation. a** Newborn C57BL/6 mice were infected with different doses of MCMV, as indicated ($n = 2$ Mock, $n = 4$ (50 PFU), $n = 5$ (200 PFU and 400 PFU), $n = 3$ (800 PFU)). **b, c** Newborn C57BL/6 mice were infected with indicated recombinant MCMV or mock-infected (**b**: $n = 5$ mice per group; **c**: $n = 2$ Mock, $n = 4$ MCMV, MULT-1 MCMV). **d** Newborn C57BL/6 mice were infected with 200 PFU (low dose, LD) or 200,000 PFU (high dose, HD) of UV-inactivated MCMV ($n = 5$ mice per group). **e, f** Newborn C57BL/6 mice were infected with 200 PFU of MCMV or mock-infected. The indicated group received i.p. ganciclovir (GCV) or immune sera on days 3, 7, and 14 p.i. (**e**: $n = 5$ mice

per group; **f**: $n = 4$ mice per group). **g** C57BL/6 female mice were infected intravenously with $2 \times 10^5$ PFU of Δm157MCMV (immunized, IM) or uninfected (non-immunized, NI), 2 weeks before mating. 2 weeks after infection, females were mated with naïve male mice. Their offspring, newborn mice, were intraperitoneally injected with 200 PFU of MCMV or mock-infected 24–36 h after birth (NI: $n = 7$ Mock, $n = 9$ MCMV; IM: $n = 10$). **a–g** On day 21 p.i. NK cell expression of Eomes was analyzed. Mean values ± SD are shown (One-way ANOVA test). Statistically significant differences are indicated (P values). Source data are provided as a Source Data file.

MCMV. In this setup, newborn mice possess maternal antibodies which protect them against infection[27]. In line with previously published data, infection was not established in these mice (Fig. S5c). Accordingly, Eomes was not downregulated in NK cells in the offspring of vaccinated mothers (Fig. 5g). These findings indicate that the level of virus replication does not play a significant role in the downregulation of Eomes in NK cells of MCMV-infected newborn mice. However, an effective vaccination approach that prevents infection can be employed as a preemptive measure.

### Conventional NK cells downregulate Eomes upon perinatal MCMV infection independently of TGF-β signaling

Conventional NK cells express and require both Eomes and T-bet for their development, whereas ILC1 cells do not express Eomes and are dependent on T-bet expression[1]. Having observed the downregulation of Eomes, we set out to determine whether the observed phenotypic change in NK1.1+ cells was due to the conversion of conventional NK cells into ILC1-like cells or to the expansion of ILC1s. Since both cell types express typical NK cell markers, to discriminate between NK cells and ILC1s, we have analyzed CD49a and CD49b expression[28,29]. As CD49a+CD49b− ILC1 numbers in the spleen are low, we have analyzed these cells in the liver, where both ILC1s and CD49a−CD49b+ NK cells are abundant (Fig. 6a). In contrast to NK cells, liver ILC1s did not upregulate CD11b and KLRG1 upon MCMV infection (Fig. 6b). To investigate if Eomes is required for the generation of terminally mature NK1.1+ cells, we have employed *Ncr1iCre Eomesf/f* (*Eomes* conditional KO or cKO) mice, in which NK cell numbers are strongly reduced due to their requirement for Eomes, while the number of CD49a+ ILC1s is unaffected[30]. Compared to infected WT mice, MCMV-infected *Eomes* cKO mice had strongly reduced numbers of terminally differentiated NK1.1+ cells (Fig. 6c). Taken together, these data demonstrate that conventional NK cells convert to terminally mature Eomeslow or Eomes− cells upon perinatal MCMV infection.

We next investigated whether terminally mature NK cells with low Eomes expression are generated in the bone marrow (Fig. 6d). A large majority of NK cells in the bone marrow expressed high levels of Eomes in both infected and control mice, indicating that NK cells mature terminally and downregulate Eomes upon migrating to the spleen and other peripheral organs. It has been shown that TGF-β can suppress Eomes expression in NK cells[3,31]. To determine if TGF-β affects levels of NK cell Eomes in MCMV infected newborn mice, we have used conditional knockout mice lacking TGF-β receptor on NK cells (Fig. 7a). The absence of TGF-β signaling did not prevent Eomes downregulation in NK cells following MCMV infection of newborn mice. Cytokine IL-12 and type I interferons have also been shown to impact Eomes levels in T cells and NK cells[32,33]. However, similar to TGF-β, we did not observe any contribution of IL-12 and IFN I to Eomes levels in NK cells following perinatal MCMV infection (Fig. 7b). Recently, it was shown that NK cell exhaustion characterized by Eomes downregulation could be induced by chronic stimulation via the activating receptor NKG2D[34]. To test the role of NKG2D in our system, we infected newborn NKG2D-knockout mice and analyzed the expression of Eomes, and observed downregulation of Eomes similar to WT mice (Fig. 7b). Stimulation with cytokines, such as the combination of IL-12 and IL-18, has been shown to mediate KLRG1 upregulation in vitro[35,36]. Therefore, we stimulated splenic NK cells isolated from naïve, 20 days old mice, with IL-12 and IL-18 and observed a strong upregulation of KLRG1 (Fig. 7c). Importantly, this stimulation also resulted in the downregulation of Eomes (Fig. 7d). Expression of IL-18 was not detected in the bone marrow of mock- and MCMV-infected mice, however, it was increased in the spleen of MCMV-infected mice on day 21 p.i. (Fig. 7e), indicating that IL-18 could drive NK cell maturation following perinatal MCMV infection. At the same time, expression of IL-12 was below the detection limit of the assay. Since IL-18 is produced upon infection of adult mice as well, we next investigated why NK cells are more responsive to IL-18

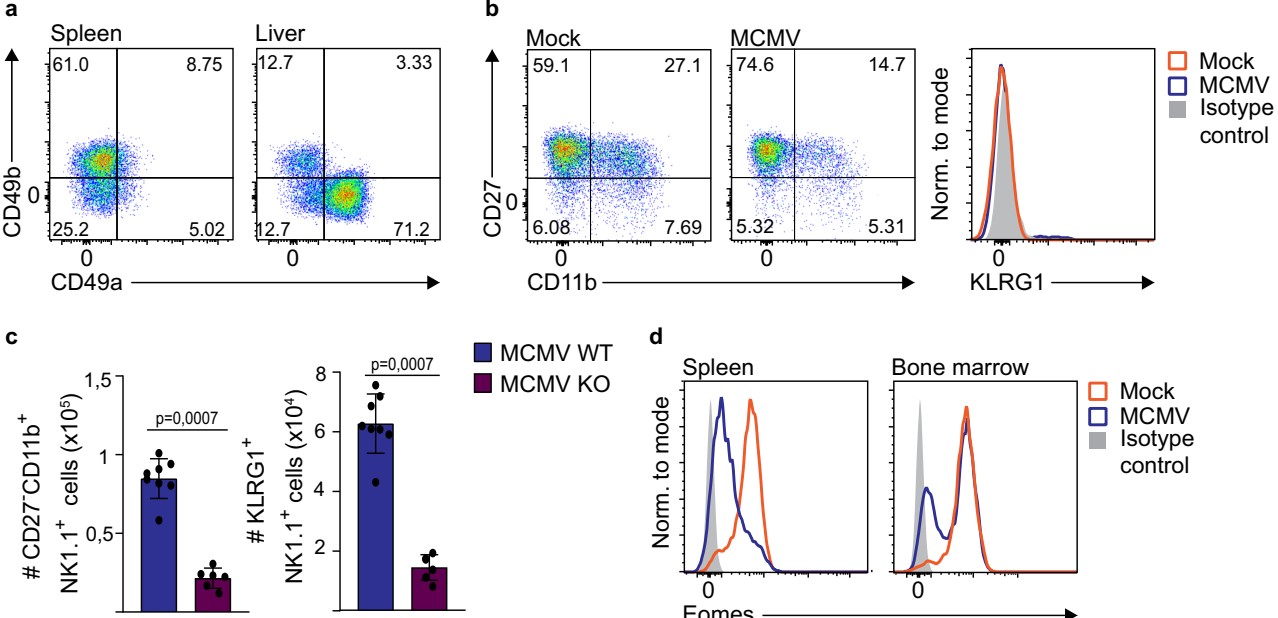

**Fig. 6 | Conventional NK cells downregulate Eomes upon perinatal MCMV infection.** Newborn mice were intraperitoneally infected with 200 PFU of MCMV or mock-infected 24–36 h after birth. **a, b** On day 21 post-infection of C56BL/6 mice, spleens and livers were harvested. **a** Expression of CD49b and CD49a was analyzed on NK1.1+ cells, N = 2. **b** Expression of CD27, CD11b and KLRG1 by ILC1s (CD49b−CD49a+) was determined by flow cytometry in the liver, N = 2. **c** On day 21 post-infection of *Ncr1iCre Eomesf/f* (KO) or *Eomesf/f* mice (WT), CD27, CD11b and

KLRG1 expression by splenic NK1.1+ cells were determined by flow cytometry. Numbers of terminally differentiated and KLRG1 expressing NK1.1+ cells are shown (n = 8 WT, n = 6 KO, Mann–Whitney two-tailed test). Mean values ± SD are shown. **d** On day 21 post-infection of C56BL/6 mice, spleens and bone marrow were harvested. Representative histograms of Eomes expression by NK cells is shown (N = 3). Statistically significant differences are indicated (P values). Source data are provided as a Source Data file.

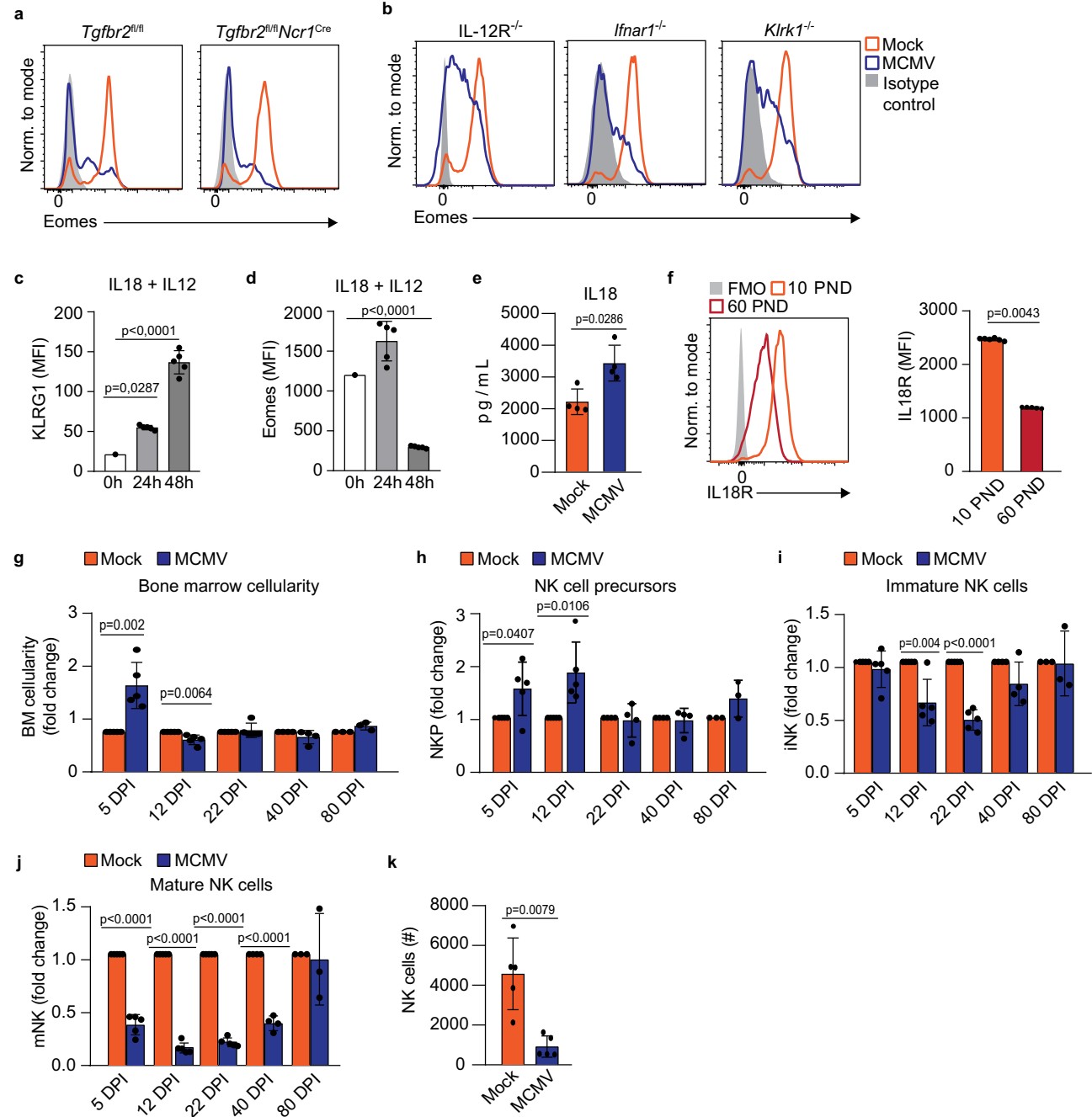

**Fig. 7 | The role of IL-18 in the downregulation of Eomes in NK cells.**
**a**, **b** Newborn mice were intraperitoneally infected with 200 PFU of MCMV or mock-infected 24–36 h after birth. **a** Expression of Eomes by NK cells lacking TGF-β receptor (*Tgfbr2^{f/f}Ncr1^{iCre}*) and control NK cells is shown (*Tgfbr2^{f/f}*) 21 days post-infection. **b** Expression of Eomes by NK cells is shown for IL-12R knockout mice (IL12R^{−/−}), type I interferon knockout mice (*Ifnar^{−/−}*) and NKG2D knockout mice (*Klrk1^{−/−}*) 21 days post-infection. Representative histograms are shown. **c**, **d** Splenic NK cells were isolated from naïve, 20 days old mice and cultured in IL-15 for 5 days. Cells were then cultured in the presence of IL-12 and IL-18 for 2 days. Quantification of KLRG1 and Eomes expression by NK cells is shown (*n* = 1 0 h, *n* = 5 24 h & 48 h; *N* = 2, One-way ANOVA test). Mean values ± SD are shown. **e** On day 21 post-infection the concentration of IL-18 was determined in the spleen (*n* = 4 mice per group, Mann–Whitney two-tailed test). Mean values ± SD are shown. **f** On postnatal days (PND) 10 and 60 the expression of IL-18R was analyzed on splenic NK cells. Representative histograms (left) and quantification (right) of IL-18R expression on NK cells are shown (*n* = 6 10 PND, *n* = 5 60 PND, Mann–Whitney two-tailed test). Mean values ± SD are shown. At indicated time points post-infection fold changes of bone marrow cellularity (**g**), NK cell precursors (**h**), and immature (**i**) and mature (**j**) NK cells were determined (*n* = 5 DPI5 & DPI12, *n* = 4 DPI22 & DPI40, *n* = 3 DPI80, Two-tailed Student's *t* test). Mean values ± SD are shown. **k** On day 21 post-infection, C56BL/6 mice were euthanized, and bone marrow cells were transferred into *Rag2^{-/-}γc^{-/-}* mice. Seven days after adoptive transfer the number of splenic NK cells was determined in *Rag2^{-/-}γc^{-/-}* mice (*n* = 5 mice per group, Mann–Whitney two-tailed test). Mean values ± SD are shown. Statistically significant differences are indicated (*P* values). Source data are provided as a Source Data file.

in young mice. We have analyzed the expression of IL-18 receptor (IL-18R) on NK cells and found that NK cells in 10 and 21 days old mice express much higher levels of IL-18R than adult mice (Figs. 7f and S6a). Therefore, NK cells in young mice are more sensitive to IL-18

stimulation than NK cells in adult mice, potentially explaining differential effects of IL-18 on NK cells in neonatal mice compared to adult mice. In addition, infection of newborn mice resulted in the downregulation of IL-18R expression in NK cells, as observed on day

21 (Fig. S6b). The low expression of IL-18R in NK cells of infected mice on day 21 p.i. was specific for the KLRG1⁺ population of NK cells.

Terminally mature NK cells undergo apoptosis and are typically replaced by fresh NK cells migrating from the bone marrow. Therefore, we expected a fast recovery of the splenic NK cell compartment following infection, as observed in adult mice (Fig. 4f–h). Since this was not the case, we hypothesized that bone marrow function was disrupted due to the infection. To determine whether NK cell production in the bone marrow is impaired by perinatal MCMV infection, we analyzed the number of NK cell progenitors, and immature and mature NK cells in the bone marrow. While the overall cellularity of bone marrow and the number of NK cell progenitors were not affected by infection by day 21 and afterward, the numbers of immature and especially mature NK cells were significantly reduced (Fig. 7g–j). To assess if the bone marrow from infected mice had a reduced ability to produce NK cells, we transferred bone marrow from MCMV-infected mice and control mice into immunodeficient mice. Lower numbers of NK cells in the spleens of recipient mice which received bone marrow cells from MCMV-infected mice were observed (Fig. 7k), showing an impaired bone marrow capacity to generate NK cells upon perinatal MCMV infection. At the same time, the generation of B and T cells was not impaired following the adoptive transfer of bone marrow cells (Fig. S6c). We did not detect MCMV genomes in bone marrow cells used for adoptive transfer and no MCMV replication in organs of recipient mice (Fig. S6d, e), implying that the state of bone marrow cells shaped in donor mice defines the generation of new NK cells. Taken together, infection-induced inflammatory response caused terminal maturation of peripheral NK cells while simultaneously causing a reduction in bone marrow capacity to generate NK cells, resulting in the maintenance of terminally mature NK cells on the periphery.

## Broad transcriptional dysregulation of NK cells following perinatal MCMV infection

We have observed that phenotypic changes in the NK1.1⁺ cell compartment are primarily restricted to the KLRG1^high population (Fig. 3b). To analyze the effects of perinatal MCMV infection at the NK subpopulation level, we sorted KLRG1⁻ NK cells from mock-infected mice (comprising > 95% of NK1.1⁺ cells in uninfected mice, Fig. 3b) and KLRG1⁻ and KLRG1⁺ cells from MCMV infected mice, and performed a bulk RNAseq analysis (Fig. 8a, b). Individual samples from the same experimental groups clustered together, indicating low inter-sample variability (Fig. 8a). Furthermore, gene ontology over-representation analysis revealed that biological processes related to lymphocyte activation, proliferation, adhesion and function were significantly impacted by the MCMV infection early in life (Fig. 8b). Surprisingly, the expression of many transcription factors was dysregulated in KLRG1⁺ NK cells (Fig. 8c), including TCF-1 and BLIMP-1, encoded by *Tcf7* and *Prdm1*, respectively, previously implicated in determining NK cell fate and in the regulation of Eomes expression[35,37–39]. Consistent with the transcriptomic data, TCF-1 was downregulated, whereas BLIMP-1 was strongly upregulated in KLRG1⁺ NK cells at the protein level (Fig. 8d, e). Taken together, these results demonstrate that perinatal MCMV infection drives a broad transcriptional dysregulation of NK cells.

## Transcriptional dysregulation of NK cells is associated with functional hyporesponsiveness

Multiple genes involved in NK cell function were transcriptionally dysregulated in KLRG1⁺ NK cells in infected mice (Fig. 9a). The expression of genes involved in cytotoxicity was enhanced in KLRG1⁺ NK cells (Fig. 9a). To verify if this transcriptional signature translates into the enhanced cytotoxic ability of these NK cells, we performed an in vitro NK cell killing assay by incubating NK cells with target RMA-S

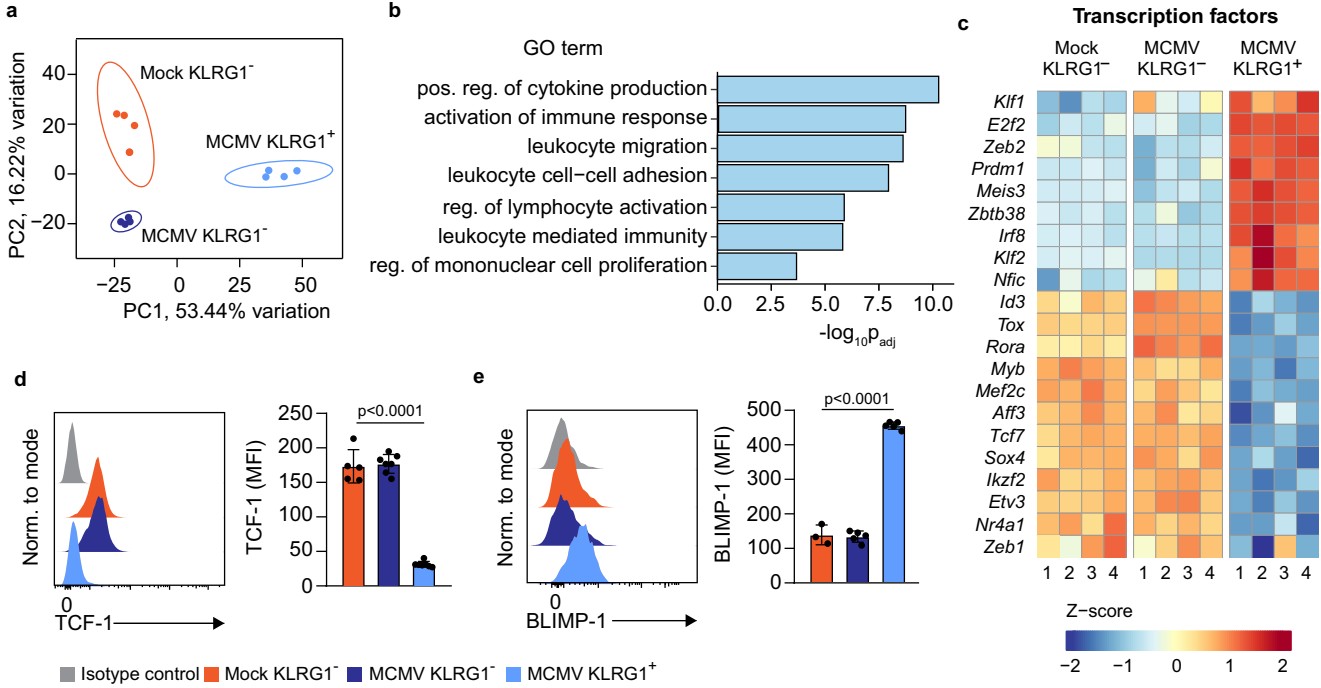

**Fig. 8 | Broad transcriptional dysregulation of NK cells following perinatal MCMV infection. a** Principal component analysis (PCA) was performed for all samples using the normalized, rlog-transformed count data, and the variability over the first (PC1) and second (PC2) principal components is shown on the x and y-axis, respectively. Ellipses around each experimental group denote a 95% confidence level. **b** Bar plots showing a selection of significantly enriched gene ontology (GO) terms in a list of genes that are deregulated in KLRG1⁺ NK cells following MCMV infection of newborn mice. **c** Heat map illustrating the differences in expression of transcription factors in KLRG1⁺ and KLRG1⁻ NK cells from MCMV-infected mice and KLRG1⁻ from mock-infected mice. **d, e** Expression of TCF-1 and BLIMP-1 was analyzed in KLRG1⁻ NK1.1⁺ cells from mock-infected mice and KLRG1⁻ and KLRG1⁺ NK1.1⁺ cells from MCMV infected mice 21 days p.i. by flow cytometry. Representative histograms and quantification of TCF-1 and BLIMP-1 expression by NK cells are shown. Mean values ± SD are shown (*n* = 5 mice per group, One-way ANOVA test was used). *P* values are shown. Source data are provided as a Source Data file.

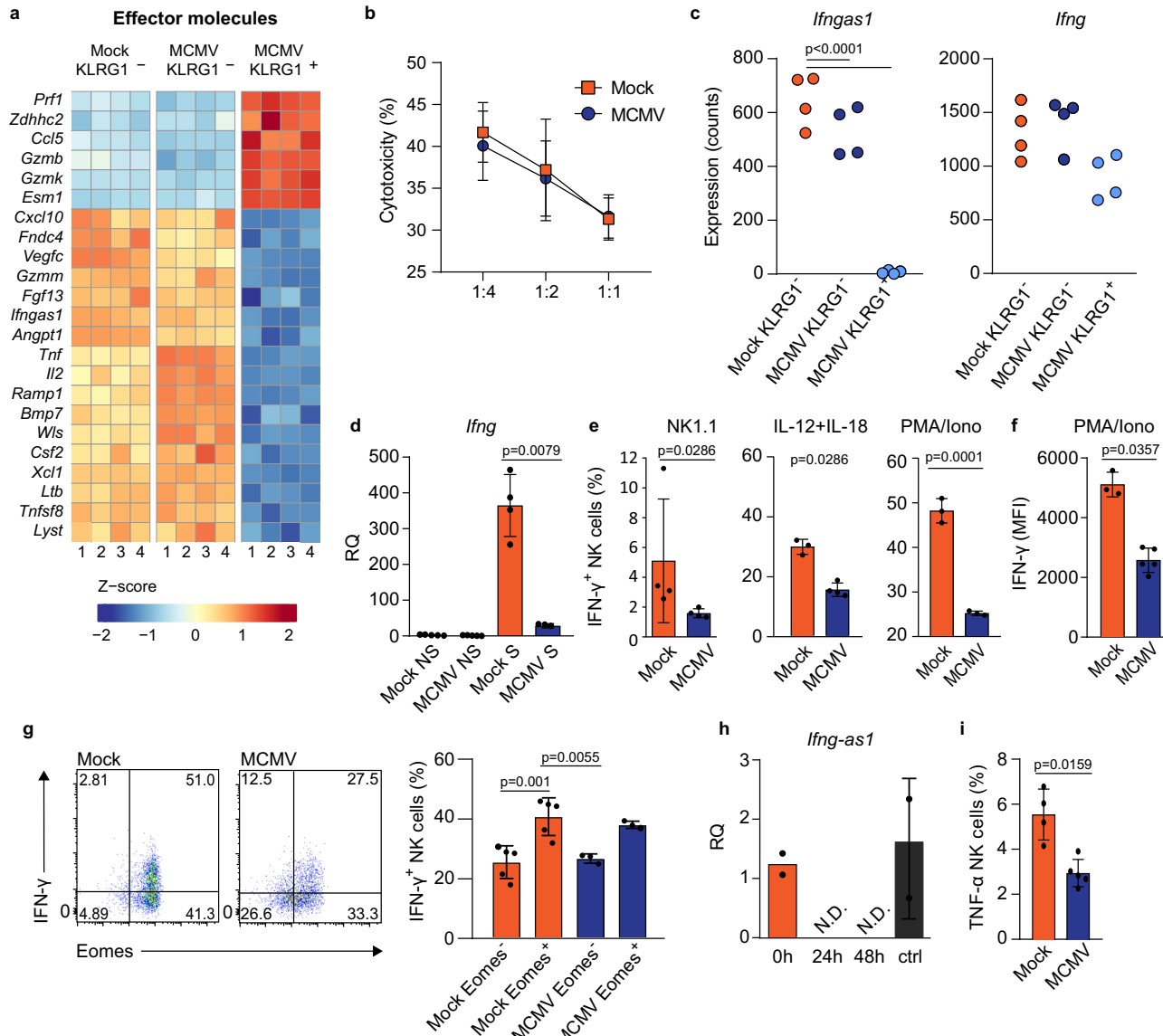

**Fig. 9 | Perinatal MCMV infection induces NK cell hyporesponsiveness.** New-born C57BL/6 mice were infected with 200 PFU of MCMV. **a** RNAseq heat map illustrating the differences in expression of effector molecules in designated cell populations. **b** Natural killer cytotoxicity assay was performed using NK cells isolated from mock- or MCMV-infected mice 21 d.p.i. incubated with RMA-S tumor cells at an effector-to-target ratios indicated in the figure ($n = 4$ mice per group). **c** The normalized numbers of reads mapped to *Ifng* and *Ifngas1* in designated NK cell types ($n = 4$ mice per group, Wald test). **d** On day 21 after infection expression of *Ifng* was determined by qPCR in non-stimulated (NS) splenic NK cells and PMA/Ionomycin stimulated (S) NK cells ($n = 5$ Mock and MCMV NS, $n = 4$ Mock S, $n = 3$ MCMV S, Mann–Whitney two-tailed test). **e** Splenocytes were stimulated with NK1.1, IL-12 and IL-18, and PMA/Ionomycin, as indicated in the figure. Quantification of IFN-γ production by NK cells is shown (NK1.1 $n = 4$, Mann–Whitney two-tailed test; IL-12 + IL-18 $n = 3$ Mock, $n = 4$ MCMV, Mann–Whitney two-tailed test; PMA/Iono

$n = 3$, Two-tailed Student's *t* test). **f** MFI values of IFN-γ production by NK cells stimulated with PMA/Ionomycin. **g** Representative plots of Eomes expression and IFN-γ production are shown (left), and quantification of IFN-γ production by Eomes[+] and Eomes[-] NK cells (right) is shown ($n = 5$ Mock, $n = 3$ MCMV, One-way ANOVA test). **h** Splenic NK cells of naïve, 20 days old mice were enriched and cultured in presence of IL-15 for 5 days, followed by culture in the presence of IL-12 and IL-18 for 2 days. Expression of *Ifng-as1* was determined by qPCR before stimulation (0 h), stimulated for 24 or 48 h, and cells kept for 48 h without IL-12 and IL-18 (ctrl) ($n = 2$ 0 h & ctrl, $n = 3$ 24 h & 48 h; $N = 3$). N.D = not detected. **i** On day 21 after infection splenocytes were stimulated with PMA/Ionomycin, and quantification of TNF-α production by NK cells is shown ($n = 4$ Mock, $n = 5$ MCMV, Mann–Whitney two-tailed test). Mean values ± SD are shown. Statistically significant differences are indicated (*P* values). Source data are provided as a Source Data file.

cells. We did not observe any significant differences in the killing capacity of NK cells from MCMV-infected mice compared to mock-infected mice (Fig. 9b). Furthermore, we performed an adoptive transfer of NK cells from infected and uninfected mice into newborn mice infected with MCMV (Fig. S7). In this case, NK cells from mock-infected mice provided slightly better protection, but the observed difference does not allow for any firm conclusions. The results suggest that increased transcriptional potential for cytotoxicity of NK cells from infected mice did not increase the killing capacity of NK cells.

Although KLRG1[+] NK cells upregulated genes mediating cytotoxic functions, genes mediating cytokine production, such as genes encoding TNF-α, XCL-1, and GM-CSF, were drastically downregulated (Fig. 9a). A prominent exception was the expression of gene encoding major NK cell cytokine, IFN-γ, which was not significantly affected (Fig. 9c). However, *Ifng-as1*, a long non-coding RNA that positively regulates IFN-γ production, was heavily downregulated in KLRG1[+] NK cells. Furthemore, expression of all *Ifng-as1* exons were equally downregulated (Fig. S8). To assess if the reduced levels of *Ifng-as1* RNA

impact the level of IFN-γ transcripts upon stimulation, we stimulated NK cells with PMA/I and performed qPCR analysis (Fig. 9d). The quantity of IFN-γ transcripts was similar in NK cells from mock- and MCMV-infected mice under steady-state conditions in accordance with RNAseq analysis. However, upon PMA/I stimulation, NK cells from mock-infected mice strongly upregulated IFN-γ transcripts, while in NK cells from MCMV-infected mice, IFN-γ transcript numbers remained low following stimulation. We have then stimulated NK cells with plate-bound NK1.1 antibody, IL-12 + IL-18, or PMA/I ex vivo and analyzed the production of IFN-γ (Fig. 9e). In all cases, fewer NK cells from MCMV-infected mice produced IFN-γ as compared to NK cells from mock-infected mice. In addition, the relative production of IFN-γ per cell was reduced in infected mice, as observed by analyzing MFI values (Fig. 9f). Furthermore, the expression of Eomes correlated with the ability of NK cells to produce IFN-γ upon stimulation, as cells with higher Eomes levels produced more IFN-γ (Fig. 9g). Since we observed that IL-12 + IL-18 stimulation results in Eomes downregulation and KLRG1 expression in NK cells, we stimulated NK cells with IL-12 + IL-18 to explore whether the same mechanism underlies the reduction of *Ifng-as1* expression. 24- and 48-h stimulation resulted in drastic downregulation of *Ifng-as1*, while cells from the same mouse that were maintained without IL-12 + IL-18 in the same period did not downregulate *Ifng*-as1 (Fig. 9h). Thus, we provide insight into the regulation of NK cells' IFN-γ expression upon viral infections. Similar to IFN-γ, the production of TNF-α by NK cells has also been reduced in infected mice (Fig. 9i). Altogether, these data indicate that infection-induced transcriptional dysregulation of NK cells leads to an impaired ability to produce cytokines.

## Discussion

NK cells rely on the expression of distinct patterns of innate immune receptors to identify and eliminate infected and tumor cells. However, different extrinsic and intrinsic factors shape the NK cell compartment in each individual. Here we employed CMV, a virus often acquired early in life in both mice and humans, and have found that MCMV infection in the perinatal period induces substantial changes in NK cells, negatively affecting multiple aspects of NK cell regulation and functionality. Contrary to adult mice, NK cells in newborn mice do not contribute to virus control, while suffering from numerous long-lasting impairments due to infection, including dysregulation of expression of multiple NK cell receptors, transcription factors, and effector molecules. These led to an impaired functional capacity of NK cells, i.e., reduced cytokine production, which was related to the downregulation of the transcription factor Eomes. This study provides the in-depth analysis of the impact of viral infections on NK cell biology in early life and provides a striking picture of the long-lasting impairment in NK cell functionality due to MCMV infection.

We have identified that mouse NK cells in the early postnatal period strongly express genes involved in the cell cycle, in accordance with the extensive proliferation of NK cells in the bone marrow, which is required to fill organs during the ontogeny[19]. Conversely, NK cells gradually acquire functional capacity over time, including the expression of Ly49 receptors. These receptors are key for NK cell function, as they control NK cell education/licensing, missing-self recognition, and direct pathogen recognition[40]. Upon MCMV infection of adult C57BL/6 mice, NK cells are activated and control MCMV via the Ly49H receptor[12,22]. In addition, NK cells proliferate vigorously in adult mice following MCMV infection, a process driven by the interaction of the NK cell Ly49H receptor and viral protein m157, resulting in the generation of adaptive NK cells[13]. These processes result in the acquisition of terminal maturation markers by NK cells. Here, we observed the activation of NK cells upon perinatal MCMV infection, however, they did not contribute to virus control. This is in accordance with previous studies showing a lack of NK cell-mediated control of MCMV infection in young and newborn mice[18,41]. An explanation for an

inefficient NK cell response to MCMV infection lies in the fact that contrary to adult mice, neonatal NK cells do not express the receptor Ly49H. Furthermore, following the resolution of perinatal infection, splenic NK cell numbers are replenished, and NK cells acquire a terminally mature phenotype. However, terminal maturation is sustained much longer after infection of newborn mice than in adult mice. Surprisingly, the transcription factor Eomes was drastically downregulated in NK cells, which is not observed upon infection of adult mice. One of the most important findings of our study was that NK cell function was impaired, which is in contrast to the increased functionality and responsiveness of adaptive NK cells generated upon MCMV infection in adult mice.

Newborn mice are more susceptible to MCMV infection, which is why we have decided to use a low dose (200 PFU) of tissue culture-derived MCMV to allow for long-term studies of consequences a viral infection can have on the immune system[42]. In our model both CD8+ T cells and CD4+ T cells are required to resolve productive MCMV infection in newborn mice[43]; however, the innate immune responses are poorly understood. Since adult C57BL/6 mice efficiently control MCMV infection via Ly49H+ NK cells, we assumed that the observed changes were due to impaired control of MCMV in newborn mice. Nevertheless, the infection of newborn mice with heavily attenuated recombinant MCMVs affected NK cells similarly, indicating that transcriptional dysregulation and impaired functions are not due to a lower capacity of newborn mice to control virus replication. Still, virus replication and a continuous inflammation and stimulation of NK cells arising from it in newborn mice was essential, as administration of low or high dose of UV-inactivated MCMV did not affect the NK cell compartment. Accordingly, administering antiviral drugs to reduce the viral burden, which do not completely block virus replication, did not prevent the infection-induced NK cell exhaustion, but the vaccination approach that prevents infection did. These findings have important implications for further research and potentially into clinical applications of antiviral drugs in the context of early-life infections. Also, these findings suggest that early replication in the first several days following infection plays a key role in shaping the NK cell compartment, as attenuated viruses cause similar disturbances in NK cells. Antiviral antibodies can decrease pathological changes in the brain of infected neonatal mice[44], and infection with attenuated recombinant MULT-1MCMV does not induce substantial inflammatory response in the brain[41], observations in sharp contrast with the impact of infection on NK cells. These differences raise an intriguing possibility that NK cell deficits caused by infection, including very mild infection, can be present even in the absence of apparent clinical disease.

Eomes and T-bet are two T-box transcription factors regulating NK cell differentiation, maturation, and function[45]. Mice lacking both Eomes and T-bet or just Eomes do not generate mature NK cells. However, NK cell numbers are not changed upon inducible deletion of Eomes in mature NK cells[32]. Certain pathological conditions, such as *Toxoplasma gondii* infection and tumors, can induce the downregulation of Eomes in NK cells, leading to the acquisition of ILC1-like properties[3,4]. In addition, NK cell Eomes downregulation has been observed upon adoptive transfer of NK cells in tumor-bearing mice[46]. In this case, the downregulation of Eomes resulted in NK cell hyporesponsiveness, which has been associated with tumor-driven NK cell proliferation. Thus, an important finding of this work is the nearly complete downregulation of Eomes following perinatal MCMV infection. While certain commonalities exist between the impact of these pathological conditions on NK cells, such as Eomes downregulation, converted NK-cells differ to some extent in all cases. We did not observe the acquisition of ILC1 properties associated with the loss of NK cells Eomes, but rather the loss of functional capacity. In the case of *Toxoplasma gondii* infection, NK cells convert to ILC1-like cells and express the prototypical marker of ILC1 cells, CD49a, which is not the case during MCMV infection of newborn mice. In addition, ILC1 cells

induced by *Toxoplasma gondii* express high levels of activating receptor DNAM-1, which is severely downregulated upon MCMV infection of newborn mice. Furthermore, in the case of tumors, TGF-β has been shown to convert NK cells to ILC1-like cells distinct from those induced by *Toxoplasma gondii* infection[3,4]. On the other side, some similarities beyond reduced Eomes expression exist. In *Toxoplasma gondii* and MCMV infections, KLRG1 is upregulated, and Eomes seems to be regulated by IL-12 + IL-18. Thus, our data expands the understanding of NK cell plasticity in pathological conditions.

NK cells appear in the first perinatal week, while ILC1 cells, lacking Eomes expression, dominate in the spleen and other organs during the embryonic and early postnatal period[17]. To rule out the possibility that MCMV infection prolonged the dominance of ILC1 cells in the spleen rather than causing loss of Eomes in NK cells, we have employed mice lacking Eomes in NCR1+ cells. We could confirm that NK cells are indeed required to generate KLRG1+ cells upon infection. Different factors have been previously identified to control Eomes levels. Human and mouse NK cells downregulate Eomes in response to enhanced TGF-β signaling[3,31], and Eomes downregulation can be induced by IL-12 and NKG2D triggering in NK cells[4,32,34]. None of these factors proved critical for Eomes downregulation in our system. However, stimulation of NK cells with IL-12 + IL-18 induced KLRG1 and resulted in Eomes downregulation, implying that some inflammatory cytokines produced upon infection can reduce Eomes expression and functional capacity of NK cells. Expression of IL-18R was found to be much higher in NK cells of young mice, potentially explaining the differential effect of IL-18 on NK cells in neonatal mice compared to adult mice. Furthermore, infection impaired the generation of NK cells in bone marrow, delaying restoration of the NK cell compartment to the state observed in naïve mice, thus explaining the prolonged maintenance of terminally mature hypofunctional NK cells in different peripheral organs. MCMV can affect bone marrow function either by infecting cells in the bone marrow, or by causing the production of inflammatory cytokines that can affect the function of bone marrow[47,48]. Our data suggest that the generation of immature and especially mature NK cells is compromised in the bone marrow. Since NK cells are not infected with MCMV, we anticipate that inflammatory conditions in the bone marrow or consequences of stromal cell infection result in unfavorable conditions for immature and mature NK cell development. However, the ultimate mechanism of reduced production of NK cells in bone marrow remains elusive. Furthermore, we have shown that the generation of B and T cells is not compromised following adoptive transfer, indicating that these effects are specific to NK cells.

Dysregulation of transcription factors resulted in a significant loss of NK cell ability to produce cytokines. Transcripts coding for IFN-γ, a major NK cell cytokine, were not differentially regulated by infection. However, long non-coding RNA (lncRNA) *Ifng-as1*, previously identified as an important positive regulator of IFN-γ production in human NK cells[49], is almost completely downregulated. A recent study showed that αCD3 or IL-12 + IL-18 stimulation induced *Ifng-as1* downregulation in murine T cells[50]. Here, we have shown that stimulation with IL-12 + IL-18 also downregulates *Ifng-as1* expression in NK cells. While this seems counterintuitive, as IL-12 + IL-18 are strong stimulators of IFN-γ expression, this phenomenon depends on timing. Stein et al. have shown that stimulation of human NK cells with IL-12 + IL-18 drive strong upregulation of *IFNG-AS1* transcription in the first several hours[49]. However, we demonstrated that prolonged stimulation has the opposite effect. Therefore, IL-12 + IL-18 stimulation can drive NK cell transcriptional dysregulation, maturation, and reduced functional capacity. The role of *Ifng-as1* in NK cells during infection and maturation processes was not studied previously. Here we have shown that loss of *Ifng-as1* is specific for KLRG1+ terminally mature NK cells induced by infection. Upon in vitro stimulation, NK cells from perinatally infected mice had a significantly reduced ability to produce IFN-γ due to the inability to increase IFN-γ-encoding transcripts.

Altogether, these data provide a paradigm of how the NK cell function can be regulated using lncRNA during maturation in an infection context. A potential explanation for the evolution of this mechanism could be that the negative regulation of IFN-γ prevents immune-mediated damage during infection, such that we have observed previously in the cerebellum of infected mice[41]. The context-dependent amelioration or disease exacerbation due to dysregulated IFN-γ expression is well-recognized[51]. Conversely, the expression of cytotoxic molecules was upregulated in NK cells but without an increase in their killing capacity. While this is surprising, a previous study has demonstrated that the terminal maturation of NK cells is associated with the upregulation of granzyme B (GzmB), which induces cell self-destruction[37]. Excess GzmB expression was assigned to the loss of NK cell TCF-1, which suppresses GzmB expression. Accordingly, we have also observed a strong downregulation of TCF-1 in terminally mature NK cells following perinatal MCMV infection. In addition, we have observed the upregulation of BLIMP-1, which is essential for GzmB upregulation[35]. Thus, an increased cytotoxic machinery probably functions as a mechanism for the self-elimination of terminally mature cells rather than a mechanism that increases the potential to eliminate target cells.

How these data relate to congenital and perinatal HCMV infection in humans remains elusive. Modeling of congenital HCMV infection in laboratory animals is challenging[52]. HCMV is strictly species-specific and cannot infect laboratory animals. Therefore, animal counterpart CMVs are employed. Mouse models are still most commonly employed to study this disease and have been instrumental in providing mechanistic insights regarding immune response and neurological deficits upon infection[53]. Since MCMV cannot pass the placenta and infect fetuses *in utero*, mice are infected perinatally in our experimental system. *In utero*, HCMV infection can occur at any time during gestation, and differences in timing can result in differences in disease progression, as infections in the first trimester are known to have worse outcomes. Here we have shown that both the infection on postnatal day 1 and postnatal day 7 caused terminal differentiation and downregulation of Eomes in NK cells. Infection of adult mice does not lead to such changes. Therefore, the impact on NK cells is not defined by specific infection time points but rather by the period of life.

Developmentally, mice are born earlier than humans. This holds true for immune system as well, with mice aged 7–10 days corresponding to human newborn[54]. Thus, newborn mouse immune system likely corresponds to human fetus. Detailed comparative studies of mouse and human NK cell development in early life are still lacking. However, there is a growing body of evidence pointing to similarities. NK cells appear in the first trimester of gestation in humans[55]. Since NK cells in mice appear postnatally, neonatal mice provide an opportunity to study the impact of infection on NK cell development during ontogeny. Importantly, direct parallels can be drawn between human and mouse NK cells in early life. Human fetal NK cells express lower levels of KIR molecules[56], an observation in line with our data on the low expression of Ly49 molecules in the early postnatal period in mice. Similarly, human fetal NK cells are suppressed by TGF-β, as shown for mouse neonatal NK cells[18]. Even though some differences in Eomes functions in mouse and human NK cells were reported recently[57], in general, the role of T-bet and Eomes seems to be very similar in human and mouse NK cells[45]. In addition, a recent publication demonstrated that *IFNG-AS1* in humans also regulates IFNG locus[49]. Altogether, similar mechanisms could operate in humans during congenital infection.

A recent study has demonstrated that *in utero* congenital HCMV infection results in expansion of more mature NK cells without apparent differences in NK cell numbers compared to uninfected infants[58], in accordance with our study. In addition, NK cells of infected congenital cases have pronouncedly developed cytotoxic machinery, and expand NKG2C+ NK cells, as we have observed in our model of perinatal infection, taking into account that there are parallels between

NKG2C[+] NK cells in humans and Ly49H[+] NK cells in mice[59]. Similar findings are reported by others[60,61]. Importantly, human NK cells of congenital cases downregulate CD7, which is indicative of IL-2 or IL-12 + IL-18 stimulation, in line with our observation that IL-12 + IL-18 stimulation seems to be a major driver of NK cell transcriptional dysregulation, maturation, and reduced functional capacity. On the other hand, Eomes expression does not differ in NK cells of congenital CMV cases compared to controls[58]. However, only a small minority of NK cells in this study expressed Eomes, irrespective of CMV status. This is in sharp contrast to other studies which show that nearly all NK cells in cord blood express Eomes[62,63]. The reason for this discrepancy is unknown, and the confirmation of data on NK cells' Eomes expression upon congenital HCMV infection is needed. We have not observed a significant impact of infection on T-bet expression by NK cells, in accordance with the data by Vaaben and colleagues. Altogether, firm parallels between our model and congenital HCMV infection with regard to the impact on NK cells seem to exist.

In summary, this study shows that perinatal infection can lead to long-lasting impairments in NK cell functions, and that during this time, the host is most likely susceptible to new infections. NK cells stand at the crossroads between innate and adaptive immunity, and their activity modulates the behavior and activity of other immune factors[64]. Our data raise an intriguing possibility that early-life NK cell dysfunction is an evolutionary adaptation that allows for the optimal formation of adaptive memory cells. Murine CMV has already proven to be an invaluable tool for studying CMV pathogenesis, and numerous findings have been confirmed in HCMV research[53]. We thus believe that our research has important implications for clinical research, not only considering the pervasiveness of CMV infection among newborns and young children but taking into account the subsequent numerous infections that are unavoidable in childhood.

## Methods

### Mice

Mice were strictly age-matched within experiments and handled in accordance with institutional and national guidelines. Newborn mice of both sexes were included in all experiments. All mice were housed and bred under specific pathogen–free conditions at the animal facility of the Faculty of Medicine, University of Rijeka where they were maintained at 22 °C in a 12-h light–dark cycle, and relative humidity (40–50%). Wild-type C57BL/6J (strain #:000664), BALB/cJ (00651), 129/SvJ (000691), Rag2[−/−]γc[−/−] (014593), Eomes[f/f] (017293) and IL12R[−/−] (003248) mice were obtained from The Jackson Laboratory. Ifnar1[−/−] mice were a kind gift from Mathias Müller[65]. Klrk[−/−] mice were generated previously[66]. Ncr1[iCre] mice were provided by V. Sexl (Vienna, Austria). Ncr1[iCre] were bred to Eomes[f/f] mice to generate Ncr1[iCre] Eomes[f/f] mice and Cre-negative littermate controls. Tgfbr2[f/f]Ncr1[iCre] mice were generated in the same manner[31]. The Animal Welfare Committee at the University of Rijeka, Faculty of Medicine and The National Ethics Committee for the Protection of Animals Used for Scientific Purposes (Ministry of Agriculture) approved all animal experiments (UP/I-322-01/18-01/30).

### Viruses and cell lines

Tissue culture-derived MCMV reconstituted from BAC pSM3fr-MCK-2fl was used in the majority of experiments[67]. In addition, recombinant MCMVs MULT-1MCMV and m74stop were used[26,68]. Virus stocks for infection of newborn and adult mice have been aliquoted and frozen at −80 °C before use. The inactivated virus was prepared by UV irradiation using UV-cross-linker. The virus was resuspended in a petri dish and placed inside the UV-cross-linker set to 5000*100 uJ/cm² in a laminar flow cabinet. The process was repeated twice. Virus stocks and organ homogenates were titrated on murine embryonic fibroblasts (MEF) using standard procedures[42]. Newborn C57BL/6 pups were infected intraperitoneally (i.p.) 24–48 h postpartum with 200 PFU of tissue cultured-passaged MCMV Smith strain. Adult, 8-week-old mice were infected i.p. with $2 \times 10^5$ PFU of MCMV. For in vitro NK cell killer assay RMA-S cells (RRID:CVCL_2180) were used as target cells. RMA-S cell line was provided by Wayne M. Yokoyama.

### Antibodies and flow cytometry

For flow cytometry, single-cell suspension of spleen, liver, lung, bone marrow, and blood leukocytes were prepared using standard protocols. In brief, mice were sacrificed, spleens harvested and homogenized, followed by erythrocyte lysis using the RBC lysis buffer. Blood samples were subjected to erythrocyte lysis. Lungs were harvested and chopped into small pieces, followed by digestion in collagenase D in 3% RPMI medium at 37 °C for 20 min with gentle agitation. After incubation, digested tissue was gently pressed through a 40-μm-pore-size nylon cell strainer using a syringe pestle and the cell strainer, and washed with 3% RPMI. The cell suspension was centrifuged for 5 min at 300 x $g$, and red blood cells were lysed with RBC lysis buffer for 5 min on ice. Lysis was blocked by adding media, after which cells were centrifuged for 5 min at 300 x $g$ and resuspended in fresh media. Hepatic leukocytes were prepared using published methods[69]. Briefly, livers were passed through a 70-μm-pore-size nylon cell strainer (BD Falcon, Franklin Lakes, NJ) and washed with 3% RPMI. Cell suspensions were then layered onto two-step discontinuous Percoll gradients (Pharmacia Fine Chemicals, Piscataway, NJ) for density separation. Hepatic leukocytes were collected after centrifugation for 30 min at 900 × $g$, followed by erythrocyte lysis. For bone marrow isolation the femur and tibia were separated by over-extending and twisting the knee joint. Any additional muscle or connective tissue attached to the femur was removed and the bones were gently opened at the knee end. The bones were placed in 0.5 mL microcentrifuge tube, previously pierced with a 18 G needle at the bottom and nested in a 1.5 ml microcentrifuge tube, knee-down and centrifuged 10,000 x $g$ for 20 s followed by erythrocyte lysis. Lysis was blocked by adding media, after which cells were centrifuged for 5 min at 300 x $g$ and resuspended in fresh media.

Before staining of lymphocytes, Fc receptors were blocked using a 2.4G2 antibody. The following antibodies were purchased from ThermoFisher: anti-mouse CD45.2 (clone 104) ef506 # 69-0454-82 (dilution 1:300), anti-mouse CD279/PD-1 (clone J43) APC # 17-9985-82 (dilution 1:100), anti-mouse CD62L (clone MEL-14) PE-Cy7 # 25-0621-82 (dilution 1:400), anti-mouse CD69 (clone H1.2F3) FITC # 11-0691-82 (dilution 1:300), anti-mouse CD103 (clone 2E7) PE # 12-1031-82 (dilution 1:200), anti-mouse IFN-γ (clone XMG1.2) FITC # 17-7311-82 (dilution 1:100), anti-mouse Bcl-2 (clone 10C4) FITC # 11-6992-42 (dilution 1:100), anti-mouse/human T-bet (clone 4B10) PE-Cy7 # 25-5825-82 (dilution 1:100), anti-mouse CD25 (clone PC61.5) PE-Cy7 # 25-0251-82 (dilution 1:400), anti-mouse CD3ε (clone 145-2C11) PerCPCy5.5 #45-0031-82 (dilution 1:100), anti-mouse CD19 (clone eBio1D3) PerCPCy5.5 #45-0193-82 (dilution 1:400), anti-mouse CD49b (clone DX5) FITC # 11-5971-82 (dilution 1:100), anti-mouse NK1.1 (clone PK136) APC # 17-5941-82 (dilution 1:100), anti-mouse NK1.1 (clone PK136) PE-eFluor 610 #61-5941-82 (dilution 1:100), anti-mouse CD127 (clone SB/199) PE # 12-1273-82 (dilution 1:100), anti-mouse CD335 (NKp46) (clone 29A1.4) FITC #11-3351-82 (dilution 1:100), anti-mouse CD335 (NKp46) (clone 29A1.4) PE-eFluor 610 # 61-3351-82 (dilution 1:100), anti-mouse CD335 (NKp46) (clone 29A1.4) PE # 12-3351-82 (dilution 1:100), anti-mouse CD27 (clone LG.7F9) PE-Cy7 # 25-0271-82 (dilution 1:100), anti-mouse CD11b (clone M1/70) FITC # 11-0112-82 (dilution 1:400), anti-mouse CD122 (clone TM-b1) FITC # 12-1222-82 (dilution 1:100), anti-mouse Ly-49H (clone 3D10) APC # 17-5886-82 (dilution 1:200), anti-mouse KLRG1 (clone 2F1) PE-eFluor 610 # 61-5893-82 (dilution 1:100), anti-mouse NKG2D (clone C7) PE-Cy7 # 25-5882-82 (dilution 1:100), anti-mouse CD94 (clone 18d3) PE # 12-0941-81 (dilution 1:100), anti-mouse Ly-49G2 (clone 4D11) FITC # 11-5781-82, anti-mouse Ly-49I (clone YLI-90) PE # A15414 (dilution 1:100), anti-mouse Ly-49A/D (clone eBio12A8 (12A8)) PE # 12-5783-81 (dilution 1:100), anti-mouse Eomes (clone Dan11Mag) PE # 12-4875-

82 (dilution 1:100), anti-mouse CD200R (clone OX-110) PE # MA1-82713 (dilution 1:100), anti-mouse CD244.1 (2B4) (clone C9.1) PE # 17-2440-82 (dilution 1:100), anti-mouse Ly6c (clone HK1.4) APC-eFluor780 # 47-5932-82 (dilution 1:100), anti-mouse CD183 (CXCR3) (clone CXCR3-173) APC # 17-1831-82 (dilution 1:100), anti-mouse CD218a (IL-18Ra) (clone P3TUNYA) PE # 12-5183-82 (dilution 1:100), anti-mouse CD226 (DNAM-1) (clone 10E5) APC # 17-2261-82 (dilution 1:400), anti-mouse NKG2A/C/E (clone 20D5) PerCP-eFluor710 # 46-5896-82 (dilution 1:100), anti-mouse CD336 (Tim3) (clone RMT3-23) PE-Cy7 # 25-5870-82 (dilution 1:100), anti-mouse MHC-I (clone 28-14-8) APC # 17-5999-82 (dilution 1:100), anti-mouse CD223 (Lag-3) (clone C9B7W) eFluor450 # 48-2231-82 (dilution 1:100), anti-mouse TIGIT (clone GIGD7) PE-Cy7 # 48-2231-82 (dilution 1:100), anti-mouse TER-119 (clone TER-119) PE # 25-5921-82 (dilution 1:100), anti-mouse CD45R (B220) (clone RA3-6B2) FITC # 11-0452-82 (dilution 1:100), anti-mouse Ki67 (clone SolA15) PerCP-eFluor710 # 46-5698-82 (dilution 1:100), anti-mouse BLIMP-1 (clone 5E7) PE # 12-9850-82 (dilution 1:100). Following antibody was purchased from BioLegend anti-mouse CD49a (clone HMα1) APC #142605 (dilution 1:100). The following antibody was purchased from Miltenyi Biotec: anti-mouse Ly49A (REA1018) VioBlue 130-117-113 F (dilution 1:50). The following antibodies were purchased from Cell Signaling: anti-mouse TCF-1 (clone C6309) PE-Cy7 #90511 S (dilution 1:400), anti-mouse TCF-1 (clone C6309) af488 #6444 S (dilution 1:400). Fixable Viability Dye (ThermoFisher) was used to exclude dead cells. Intracellular staining, permeabilization, and fixation of cells were done using the Fixation/Permeabilization kit (Thermo Fisher Scientific). Gating strategy for NK and ILC1 cells is shown in Fig. S9. Stimulation of NK cells was done in the presence of Brefeldin A (1,000×; Thermo Fisher Scientific). Intranuclear staining, permeabilization, and fixation of cells were done with the FoxP3 staining buffer set (Thermo Fisher Scientific). All samples were acquired using FACSAriaIIu, and data were analyzed using FlowJo v10 (Tree Star) software.

### In vivo treatments and adoptive transfers

To investigate the impact of NK cells on the control of MCMV, NK cell depletion was carried out by injecting αNK1.1 antibody (PK136, BioX-Cell) intraperitoneally (i.p.) twice a week over a period of 21 days. To evaluate the role of NK cells in MCMV control following infection on postnatal day 1, 7, and 14, mice were depleted of NK cells by administering depleting αNK1.1 antibody one day before and one day after infection. For mice aged ≤10 days, a dose of 50 µg of antibody in 50 µL of PBS was injected. For mice aged 11–21 days, a dose of 100 µg of antibody in 100 µL of PBS was injected. For adoptive transfer of NK cells, in vivo CD4 and CD8 T cell depletion was performed by i.p. injection of 100 µg of anti-CD4 (GK1.5, BioXCell) and anti-CD8 antibody (2.43, BioXCell) per mouse. Antibodies were administered the day before donor mice were sacrificed. 4 µg/g of poly-IC (InvivoGen) was administered intraperitoneally on postnatal days 1, 6 and 13. MCMV-infected or mock-infected mice were treated daily with 60 µg/g GCV (Cheplapharm) and with immune sera every 4 days post-infection.

For the bone marrow transfer experiments, mock- and MCMV-infected mice were euthanized 21 days post-infection, and bone marrow cells were isolated and transferred into gender-matched Rag2$^{-/-}$γc$^{-/-}$ recipient mice. Recipient mice were euthanized 7 days after adoptive transfer. For assessment of the protective capacity of transplacentally transferred antibodies, C57BL/6 female mice were immunized intravenously with $2 \times 10^5$ PFU of Δm157MCMV or left uninfected. 2 weeks after infection, female mice were mated with naïve male mice. Their offspring, newborn mice were intraperitoneally injected with 200 PFU of MCMV or mock- infected 24–36 h after birth.

### RNA sample preparation, quality control and RNA sequencing

To determine changes in NK cells in the postnatal period, splenocytes were isolated from spleens of naive mice at postnatal days 7, 14, 21 and 60 (NK-TimeCourse). To determine the impact of infection on NK cells, splenocytes were isolated from spleens of naive or MCMV-infected mice at day 21 p.i. (NK-KLRG1). Following isolation, splenocytes were labeled with anti-CD45, anti-CD3, anti-CD19, anti-NK1.1, and anti-DX5 antibodies, and NK cells defined as CD45 + , CD3-, CD19-, NK1.1 + , and DX5+ cells, were separated from the mixture using fluorescence-activated cell sorting on FACSAriaIIu using a 70-µm nozzle. Sort purity was determined by sorting an aliquot of cells into 10% RPMI and then immediately reanalyzing the sorted aliquot by flow cytometry. In general, we achieved sort purities of >98%. NK cells for RNA isolation were sorted directly into RLT lysis buffer (Qiagen) and their total RNA was isolated using RNeasy Micro Kit (Qiagen), according to the manufacturer's recommendations. The quality and quantity of isolated total RNA were then estimated on Agilent Bioanalyzer 2100 using the Agilent RNA 6000 Nano Kit, and only samples with RNA integrity number values higher than 9.0 were used. Before library generation, RNA was subjected to DNase I digestion (Thermo Fisher Scientific) followed by RNeasy MinElute column clean up (Qiagen). RNA-seq libraries were generated using the SMARTSeq v4 Ultra Low Input RNA Kit (Clontech Laboratories) as per the manufacturer's recommendations. From cDNA, final libraries were generated using the Nextera XT DNA Library Preparation Kit (Illumina). Concentrations of the final libraries were measured with a Qubit 2.0 Fluorometer (Thermo Fisher Scientific), and fragment length distribution was analyzed with the DNA High Sensitivity Chip on an Agilent 2100 Bioanalyzer (Agilent Technologies). All samples were normalized to 2 nM and pooled at equimolar concentrations. The library pool was sequenced on the NextSeq500 (Illumina) in a single 1 × 76–bp run, producing 20.0–23.8 M reads per sample from a total of twelve mRNA-seq libraries for the NK-KLRG1 RNASeq experiment. For the NK-TimeCourse RNASeq experiment, the library pool was sequenced on the NextSeq500 (Illumina) in a single 1 × 72-bp run, producing 15.6–24.0 M reads per sample from 16 mRNA-Seq libraries. For both experiments and before downstream data processing and analysis, adapter sequences were hard-clipped from raw sequencing reads as part of the bcl2fastq pipeline (version 2.20.0.422).

Computing environment, gene expression estimation, differential expression analysis, gene ontology over-representation analysis (GO-ORA) and RNA-Seq results visualization are described in detail Supplementary methods.

### qPCR

RNA was extracted from sorted or enriched NK1.1$^+$ cells using an RNA extraction kit (NucleoSpin TriPrep; MN), and cDNA was generated using a reverse transcription kit (iScript Reverse Transcription Supermix for RT-qPCR; Bio-Rad). The mRNA expression was examined by qPCR with a 7500 Fast Real Time PCR machine (ABI). TaqMan assays were used to quantify the expression of *Ifng* (Mm01168134_m1) and *Ifngas1* (Mm01161206_m1) according to manufacturers recomendations. Relative mRNA expression was normalized by quantification of *Gapdh* (Mm05724508_g1) RNA in each sample. To quantify MCMV genome copies, lungs and bone marrow were collected and total DNA was extracted using NucleoSpin TriPrep (MN). Quantification of MCMV and mouse genome copies was performed as described previously (Simon CO[70]). Viral *gB* and mouse *Pthrp* sequences were assayed in technical duplicates using 5 µl of 50 ng sample DNA per reaction. Serial dilutions of $10^1$–$10^6$ copies per reaction of the pDrive_gB_PTHrP_Tdy plasmid were used to generate standard curves for both qPCR reactions. Quantitative PCR was performed using Fast Plus EvaGreen qPCR master mix (Biotium) in a 7500 Fast Real Time PCR (Applied Biosystems). Cycling conditions were as follows: enzyme activation, 2 min 95 °C, followed by 50 cycles of denaturation for 10 s at 95 °C, annealing for 20 s at 56 °C and extension for 30 s at 72 °C. Primer sequences that were used: *Pthrp* forward 5'-ggtatctgccctcatcgtctg-3' and reverse 5'-cgtttcttcctccaccatctg-3', *gB* forward 5'-gcagtctagtcgcttctgc-3' and reverse 5'-aaggcgtggactagcgataa-3'.

## Cytokine analysis

The concentration of cytokines was analyzed using a commercially available multiplex panel mouse cytokine assay kit (MTH17MAG-47K) (Merck Millipore, Darmstadt, Germany). This Luminex multiplex ELISA-based immunoassay contains fluorescent dyed beads coated with antibodies specific for cytokines (Luminex, Austin, TX, USA). The concentration was measured using a MAGPIX array reader (Luminex) that quantifies cytokines using very small sample volumes. Cytokine concentrations were calculated using standard curves using software provided by the manufacturer (Luminex Manager Sofware).

## In vitro stimulation and killer assay

Mice were sacrificed, and splenocytes were isolated using a standard protocol. $2 \times 10^6$ cells were then incubated for 6 h with PMA/Ionomycin, cytokines IL-12 and IL-18, or αNK1.1 in RPMI 1640 (PAN-Biotech) supplemented with 10% FCS (PAN-Biotech), Brefeldin A (Invitrogen) and Monensin (Invitrogen) at 37 °C. Splenic NK cells were isolated from naïve, 20 days old mice and were enriched using NK cell isolation kit (Miltenyi Biotec) and cultured with 50 ng/mL IL-15 for 5 days in RPMI 1640 (10% FCS) at 37 °C and 5% $CO_2$. Cells were then cultured in the presence of IL-12 (15 ng/mL) and IL-18 (50 ng/mL) for 2 days. For the in vitro killer assay, mice were sacrificed, and splenocytes were isolated. RMA-S cell line was cultured in RPMI 1640 (10% FCS) at 37 °C and 5% $CO_2$. Splenocytes were co-incubated with CFSE-labeled RMA-S cells for 4 h in 4:1, 2:1, and 1:1 effector-to-target ratios at 37 °C. After co-incubations, target cells were identified as CFSE positive, and viability was determined using TO-PRO-3 (ThermoFisher). NK cell cytotoxicity was calculated using the following formula: [(% TO-PRO-3 + CFSE+ cell-specific lysis − % TO-PRO-3 + CFSE+ cell spontaneous lysis)/(100 − % TO-PRO-3 + CFSE cell spontaneous lysis)] × 100[71].

## Statistical analysis

Statistical analysis for RNAseq is described in Supplementary methods. For other analyses, raw data values were inspected for normal distribution using the Prism 5 software (GraphPad Software Inc.) using available tests for normality. The selection of the appropriate test was dependent on the number of animals per group and data distributions (two-tailed Mann–Whitney test, Student's $t$ test or One-way ANOVA test). A level of $p < 0.05$ was considered to be statistically significant.

## Reporting summary

Further information on research design is available in the Nature Portfolio Reporting Summary linked to this article.

## Data availability

The RNA sequencing data data generated in this study have been deposited in the European nucleotide archive under accession code PRJEB64583. Datasets generated during and/or analyzed in this study are available from the corresponding author upon request. Source data are provided with this paper.

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

## Acknowledgements

This work has been fully supported by the Croatian Science Foundation under the project IP-2018-01-4435 (I.B.) and the grant "Strengthening the capacity of CerVirVac for research in virus immunology and vaccinology" [KK.01.1.1.01.0006] granted to the Scientific Centre of Excellence for Virus Immunology and Vaccines and co-financed by the European Regional Development Fund (S.J.). This work was also supported by NIH grant R01-AI131680 (W.M.Y.). We thank Tihana Tršan, Edvard Ražić, Dijana Rumora, Ante Miše, Mihaela Gašparević, Cristina Paulović and Antonija Šarlija for the excellent technical and administrative support. We thank Luka Traven and Jurica Arapović for critical readings of the manuscript, and Felix M. Wensveen for helpful discussions.

## Author contributions

I.B. and S.J. conceived the study; I.B., C.R., M.P.M., A.M., L.H., E.P., A.L.B., D.I., I.V. and V.J.L. performed the experiments; I.B., C.R., B.L. and M.Š. analyzed the data; B.A. provided critical materials. I.B. wrote the manuscript; S.J., VJL, C.R., B.L., A.K. and W.M.Y. reviewed and edited the manuscript. All authors have read and agreed to the published version of the manuscript.

## Competing interests

The authors declare no competing interests.
