## [Peer Review File · Nature Communications]

Reviewers' Comments:

Reviewer #1:

Remarks to the Author:

The manuscript submitted by Rožmanić et al provide an in depth characterization of the impact of perinatal MCMV infection on NK cells in mice. The data indicate that MCMV induces the differentiation of NK cells with an altered transcriptional profile and impaired effector functions as compared to the NK cell response in adult animals. Data also indicate that perinatal MCMV infection is associated with a reduced generation of NK cells in the bone marrow. Overall, the experimental approach is well conceived and coherently addresses the aim of the study. The results are original and provide novel insight in the immunobiology of MCMV infection in early life. However, the study suffers a major limitation, the uncertainty about its translatability to human CMV infection in early life.

The authors rightly indicate that current knowledge on the role of NK cells in congenital or perinatal human CMV infection is limited and they state that perinatal MCMV infection represents an appropriate model for human infection. This statement is quite general and it is unclear whether it applies to the research question. NK cells develop in the first trimester of gestation in humans, much earlier than their perinatal development in mice. So, the NK cells responding to perinatal CMV infection are at very different stage of their development in mice and humans and this is likely to have important implications for the response and the role of this cell subset. This point is not discussed by the authors. The authors also fail to cite publications indicating that congenital and perinatal human CMV infection induces the differentiation of effector NK cells (for example Loyola et al, 2012 and 2015; Vaaben et al, 2022). Vaaben et al. actually provided data suggesting no influence of congenital CMV infection on the expression of T-bet or Eomes by NK cells. These data seem in contradiction with the results reported by Rožmanić et al. Authors should cite and discuss published literature on NK cell response to human CMV infection in early life. If they consider that published data do not point to a different impact of CMV infection on mouse and human NK cells, they should provide some human data supporting their conclusions.

Some other components of the manuscript may be further consolidated:

- The authors state that NK cells do not contribute to the control of perinatal MCMV infection in their model. The data (Figure 1B) on which this strong statement is based appear limited. The experiment includes a small number of animals and a trend towards higher viral load is actually observed in NK cell depleted animals. The authors also state that newborn mice have a reduced capacity to control MCMV infection as compared to adult mice. Yet, no viral load was detectable at d21 post-infection in young mice. This indicates a capacity of neonatal mice to control viral replication at this infectious dose. Authors could comment on the possible mechanisms underlying viral control.
- The reduced production of NK cells in the bone marrow following perinatal infection is intriguing. Are NK cells the only affected lineage? It would be important to know whether other lineages are affected as well.
- In the bone marrow transfer experiment (Figure 5J), was MCMV transferred too and were recipients mice infected following transfer? This information would help interpret the presented data.

Reviewer #2:

Remarks to the Author:

The study by Rožmanić et al reports interesting and novel findings regarding how viral infection early in life impacts NK cell development and antiviral immunity. The authors show that viral infection in young mice had a profound impact on NK cell function and maturation, associated with down-regulation of key transcription factor Eomes in terminally differentiated NK cells, with altered dysregulation of multiple transcription factors and cytokine expression. They provide evidence for the requirement of limited virus replication and, possibly, cytokines in this process.

Major Comments:

This is a very interesting study and the concept that infection early in life detrimentally alters NK cell development is important. The majority of conclusions from the study are well-supported by the data. The aspect that is less well-developed is an understanding of the virus-induced factors that induce NK cell defects *in vivo*, and when:

In vitro-derived data demonstrating that the combination of IL-12 and IL-18 drive Eomes down-regulation is convincing. However, the authors only show IL-18 is up-regulated on day 21 post-infection in spleens upon MCMV infection. Is IL-12 also up-regulated? The kinetics of up-regulation of IL-18 at day 21 fits with KLRG1 induction at this time, but data in Figure 4, including experiments with therapeutic administration of antivirals, implies that early not late low-level virus replication is required for the NK cell dysfunction. Does early virus replication also lead to cytokine induction at these later time-points?

Why do the authors believe that Eomes is only down-regulated in mature NK cells during MCMV infection despite mature and immature NK cells expressing comparable Eomes levels (as per Fig. 3C)? Do they express altered expression of IL-12R/IL-18Rs as compared to immature cells? Does this alter between young and old mice? Or is there a more complex interaction related to difference between mature/immature NK cells in young versus old mice? Some explanation would improve the understanding of mechanism.

Fig 5C shows that Eomes expression by NK cells is required for accumulation of CD11b+CD27⁻ NK cells. Paradoxically, Fig. 2C demonstrates increased CD11b+CD27⁻ NK cell accumulation after MCMV infection and Fig 3 reports Eomes down-regulation upon MCMV infection. How do the authors explain these apparent conflicting data? Also, data presented in Fig 5C doesn't support the conclusion that 'these data demonstrate that conventional NK cells, not ILC1s, convert to terminally mature Eomes^{low} or Eomes⁻ cells upon perinatal MCMV infection' and perhaps would be better suited elsewhere in the manuscript.

Minor Comments:

- Is UV virus not inflammatory? Does it induce cytokines *in vivo*? If not, presumably this is due to too few virions in preparations to induce measurable cytokine responses *in vivo*. This is worth showing/explaining.
- Fig 4E&F. These results, also with Fig. 4B&C, imply that early low-level replication is required for Eomes down-regulation. This is an important implication of the data and is worth stating.
- Fig 1. Please show that NK1.1 deletion effectively led to deletion of NK cells to prove the point that this cell type is ineffective in early virus control.
- Line 188, please change 'is' to 'are'
- Line 224, please introduce MCMV-MULT-1
- Presumably, the authors will provide the full datasets related to Fig 6?
- Statistics, multi-group analysis should be performed when there are more than two groups.

Reviewer #3:

Remarks to the Author:

In this interesting paper by Rožmanić and colleagues, the impact of CMV on NK cells is examined. The particularly intriguing aspect of the paper is that it is examined in the context of a neonatal/perinatal infection model. This greatly adds to the value and interest of the paper. So many (nearly all) papers exploring the *in vivo* biology of NK/CMV interactions have done so in the context of a transplant/oncology paradigm, and yet the biggest impact of CMV on human health is congenital infection. This makes the paper particularly valuable, especially for pediatric physician-scientists.

The model system the authors use is murine CMV (MCMV). Since MCMV does not cross the placenta, a neonatal infection model is used that mimics the pathologies of human congenital CMV. It is not perfect, but its relevance is convincing.

The authors found that MCMV infection of newborn mice severely impacted the NK function.

The paper is valuable, the experiments well designed and well-controlled, and the conclusions supported by the data. The manuscript advances our understanding of the impact of congenital (presumably, recognizing the intrinsic challenges of a post-natal challenge model) CMV infection on NK cell biology.

A few suggestions are offered for ways the manuscript can be enhanced/clarified.

1. The infection had to be early in the neonatal period.

In C57BL/6 mice, the NK cell activating receptor Ly49H provides control of MCMV infection. NK cells do not express the Ly49H receptor in newborn mice, explaining the lack of NK cell dependent control of MCMV in the early period of life. Can the authors provide:

a. A clearer elucidation of when this deficit is overcome, i.e., what day of life?

b. A clearer elucidation of how it is overcome, namely, the mechanism that transitions the neonatal mouse to a more "adult" immunophenotype?

c. Some perspective on other mouse strains. In other words, is this all unique to C57BL/6? What about BALB/c mice? This is not a trivial point with respect to the translational implications. If this is so highly strain specific that the mechanisms don't translate to different mouse strains, what is the "take-home" message for human health in the context of cCMV?

2. The impact was evident even if replication-impaired viral mutants were used.

The mutants described were MULT-1MCMV and Δ m74MCMV. The use of the UV inactivated virus that abrogates the effect completely is a nice control (Fig. 4D). One of these fails to make a gH/gL/gO complex. Other CMV systems suggest that gH is absolutely essential, but gO is dispensable. The authors should explain how any viral DNA replication can take place with a gH knockout. Isn't this incompatible with replication of any sort?

3. Infection modified Eomes transcription, resulting in impaired NK cell function.

The authors do a good job of elucidating the impact of Eomes on NK cells in this mouse system. The key point that they don't comment on in the discussion is human Eomes. Is the role of Eomes in humans similar to mice? Does perinatal/congenital infection have the same impact on NK cells in other mammalian systems, or with other pathogens (they do reference the Park paper, which might merit more comments - was the Park paper's toxoplasmosis work performed in C57/BL6 mice?

4. There was a major impact on a long non-coding RNA, IFNG-AS1, which appeared to mediate an effect on cytokine production. Figure 7D shows the qPCR data. Call me old-fashioned, but this reviewer would like to see this corroborated by Northern blot, to confirm that the altered PCR signal wasn't an artifact of changes in splicing or alternative promoter use of an upstream promoter. More to the point, what would be the mechanisms of this impact on transcription? The authors point out that it's a noncoding RNA. Is it even a pol II transcript? Is IFN gamma RNA the right control? How does IFNG-AS1 impact NK cell maturation? Is IFNG-AS1 conserved in humans? Again, the relevance to human cCMV biology is important to underscore here, since this mouse model is not truly a congenital infection model. Did, for example, the authors see any of the types of changes that are observed following early life and/or congenital CMV infections in infants (Vaaben AV, Levan J, Nguyen CBT, et al, doi:10.1093/infdis/jiac307).

5. There was an impact on the ability of the bone marrow to generate new NK cells. Figure 5 K-N

is convincing on this point. The phenotype is transferable by adoptive transfer, which is interesting. Do the authors have a hypothesis as to why there is a reduction in bone marrow capacity to generate NK cells?

6. Antiviral treatment was unable to prevent these effects. This extends to immune globulin. Interestingly immune globulin has been shown to abrogate some of the developmental defects in these neonatal mice, including cerebellar hypoplasia, by these authors. The authors seemed to treat with immune sera and/or ganciclovir after viral challenge. The more interesting experiment would have been to treat BEFORE viral challenge. This would have obvious implications for CMV vaccines for congenital CMV infection.

Rožmanić et al.
RESPONSE TO REVIEWERS

Dear Dr. Mason and dear reviewers,

We would first like to thank you for the opportunity to get our data peer-reviewed for Nature Communications and for the time and effort invested in reviewing our manuscript. We thank editor for not insisting on validation of our data on clinical samples of congenital HCMV cases, as this would require several years of additional work. We thank the reviewers for their overall positive comments and their suggestions for improving the manuscript. In the revised manuscript, we have addressed all of the concerns raised by reviewers and included the results of dozens of new experiments that further strengthen our initial findings. Below we provide point-by-point response to reviewers (in blue), and we hope that our manuscript is now suitable for publication in Nature Communications.

Reviewer #1 (Remarks to the Author):

The manuscript submitted by Rožmanić et al provide an in depth characterization of the impact of perinatal MCMV infection on NK cells in mice. The data indicate that MCMV induces the differentiation of NK cells with an altered transcriptional profile and impaired effector functions as compared to the NK cell response in adult animals. Data also indicate that perinatal MCMV infection is associated with a reduced generation of NK cells in the bone marrow. Overall, the experimental approach is well conceived and coherently addresses the aim of the study. The results are original and provide novel insight in the immunobiology of MCMV infection in early life. However, the study suffers a major limitation, the uncertainty about its translatability to human CMV infection in early life.

The authors rightly indicate that current knowledge on the role of NK cells in congenital or perinatal human CMV infection is limited and they state that perinatal MCMV infection represents an appropriate model for human infection. This statement is quite general and it is unclear whether it applies to the research question. NK cells develop in the first trimester of gestation in humans, much earlier than their perinatal development in mice. So, the NK cells responding to perinatal CMV infection are at very different stage of their development in mice and humans and this is likely to have important implications for the response and the role of this cell subset. This point is not discussed by the authors.

The authors also fail to cite publications indicating that congenital and perinatal human CMV infection induces the differentiation of effector NK cells (for example Loyola et al, 2012 and 2015; Vaaben et al, 2022). Vaaben et al. actually provided data suggesting no influence of congenital CMV infection on the expression of T-bet or Eomes by NK cells. These data seem in contradiction with the results reported by Rožmanić et al. Authors should cite and discuss published literature on NK cell response to human CMV infection in early life. If they consider that published data do not point to a different impact of CMV infection on mouse and human NK cells, they should provide some human data supporting their conclusions.

We agree with the reviewer that relating our findings to human infection is not easy and should be addressed adequately in the discussion. We have introduced a new paragraph in the discussion section of the revised manuscript to discuss these issues.

Modeling congenital HCMV infection in laboratory animals is challenging (Moulden et al. 2021). HCMV is strictly species-specific and cannot infect laboratory animals. Therefore, animal counterpart CMVs are employed to study the pathogenesis of CMV infection. While some animal models, such as guinea pigs and non-human primates, offer the advantage of transplacental virus transmission, they have significant disadvantages for mechanistic studies due to lack of genetic models, availability of reagents, high costs, and ethical concerns. Thus, mouse models are still the most commonly employed experimental approaches to study CMV disease and have been instrumental in providing mechanistic insights regarding immune response and neurological deficits upon infection (Reddehase and Lemmermann 2018).

Since MCMV cannot pass the placenta and infect fetuses in utero, mice are infected perinatally in our experimental system. While such an approach is not ideal, we have previously utilized this kind of infection model to study congenital MCMV infection and its consequences on brain development (Koontz et al. 2008, Cekinović et al. 2008, Kveštak et al. 2021), and have demonstrated that perinatal infection of mice recapitulates pathologies observed in human infants following congenital HCMV infection. As in every experimental model, one cannot account for every variation. We have used initially only one time-point, whereas, in humans, the infection can occur at any time during pregnancy, with worse outcomes if it occurs earlier. We have now added a later time point (7th postnatal day) for the infection of mice to model the CMV disease more comprehensively. Similarly to infection on postnatal day 1, infection of mice on post-natal day 7 caused terminal differentiation and downregulation of Eomes in NK cells (Fig. S4). As was shown in the manuscript, infection of adult mice does not lead to such changes. Therefore, the impact on NK cells is not defined by specific infection time points but rather by a period of life.

Detailed comparative studies of mouse and human NK cell development in early life are still lacking. However, there is a growing body of evidence pointing to similarities. As mentioned by this reviewer,

NK cells appear in the first trimester of gestation in humans (Uksila et al. 1983). Since NK cells in mice appear postnatally, this argues in favor of using neonatal mice to study the impact of infection on NK cell development during ontogeny. Importantly, direct parallels between human and mouse NK cells in early life exist. Human fetal NK cells express lower levels of KIR molecules (Ivarsson et al. 2013), an observation in line with our data on the low expression of Ly49 molecules in neonatal mice. Similarly, TGF- β suppresses human fetal NK cells; the same is true for mouse neonatal NK cells (Marcoe et al. 2012). In addition, a recent publication demonstrated that IFNG-AS1 in humans also regulates the IFNG locus (Stein et al. 2019). Therefore, similar mechanisms could operate in humans upon infection.

In utero congenital HCMV infection results in signs of maturation and expansion of more mature NK cells without apparent differences in NK cell numbers compared to uninfected infants (Vaaben et al. 2022), in accordance with our study. In addition, NK cells of infected congenital cases have increased cytotoxic machinery and expand NKG2C⁺ NK cells, as we have observed in our model of perinatal infection, taking into account that there are parallels between NKG2C⁺ NK cells in humans and Ly49H⁺ NK cells in mice. Similar findings are reported by others (Noyola et al. 2015, Noyola et al. 2012). Importantly, human NK cells of congenital cases downregulate CD7, indicative of IL-2 or IL-12+IL-18 stimulation. As shown in the revised manuscript, IL-12+IL-18 stimulation seems to be a major driver of NK cell transcriptional dysregulation, maturation and reduced functional capacity in our model.

On the other hand, Eomes expression does not differ in NK cells of congenital CMV cases compared to controls (Vaaben et al. 2022). Vaaben and colleagues provide data that only a small minority of NK cells express Eomes, unrelated to CMV status. This finding is in stark contrast to other published data, such as that of Collins and colleagues (Collins et al. 2017), who clearly show widespread expression of Eomes by NK cells from the cord blood of elective abortions. In addition to Collins et al., others have shown that human cord blood NK cells express Eomes, e.g., (Bennstein et al. 2020). Flow cytometry data in Vaaben et al. does not contain any positive staining for Eomes so one cannot exclude that the absence of Eomes staining is not caused by faulty reagents failing to detect Eomes. Our data parallels that of Vaaben and colleagues on minor upregulation of T-Bet where staining is clearly positive. *Altogether, firm parallels exist between our model and congenital HCMV infection regarding its impact on NK cells.* However, studies are needed to determine the extent to which human and mouse development in the context of infection overlap, including the kinetics of response. Our study is the necessary first step as it is certainly easier to set the foundations in animal models to enable more targeted research in humans.

Some other components of the manuscript may be further consolidated:

- The authors state that NK cells do not contribute to the control of perinatal MCMV infection in their model. The data (Figure 1B) on which this strong statement is based appear limited. The experiment includes a small number of animals and a trend towards higher viral load is actually observed in NK cell depleted animals.

We agree with the reviewer's comment that it could be concluded that there is a trend towards better control of MCMV in mice that were not depleted of NK cells. As stated in the manuscript, this experiment was performed twice, and there were no statistically significant differences in both cases. We have now performed an additional experiment and included new data for day 7 in the revised manuscript (Fig. 2A). We have also included data for the liver, lungs and brain to emphasize further the lack of NK cell-mediated control following infection of newborn mice (Fig. S2). Importantly, we have performed additional experiments to elucidate when the control of MCMV by NK cells starts in young mice (Fig. 1E, Fig. S1). This data further strengthens our conclusions that neonatal NK cells cannot control MCMV. Furthermore, early in their development, mice do not express Ly49H, a strong activating receptor of NK cells. Mice lacking Ly49H receptor are deemed MCMV-sensitive as their NK cells control the infection more poorly in various organs than those expressing Ly49H. Therefore, the kinetics of Ly49H acquisition on NK cells corroborates our findings. In addition, earlier studies have shown that adoptive transfer of Ly49H⁺ NK cells reduces MCMV burden in neonatal mice (Sun, Beilke and Lanier 2009).

The authors also state that newborn mice have a reduced capacity to control MCMV infection as compared to adult mice. Yet, no viral load was detectable at d21 post-infection in young mice. This indicates a capacity of neonatal mice to control viral replication at this infectious dose. Authors could comment on the possible mechanisms underlying viral control.

We are using a low dose (200 PFU) of tissue culture-derived MCMV; therefore, newborn mice efficiently control infection (Brizic et al. 2022). LD-50 for newborn mice is ~5-fold higher (1000 PFU). Adult C57BL/6 mice are typically infected with 10 000 – 100 000 PFU. However, adult C57BL/6 mice survive challenges with up to 10⁷ PFU of MCMV, which is a maximum that can be achieved by infecting mice with cell culture-derived stocks of MCMV. Thus, newborn mice are more susceptible to MCMV infection. While the innate mechanisms controlling MCMV infection in neonatal mice are not well

Rožmanić et al.
RESPONSE TO REVIEWERS

studied, we have previously shown that both CD8⁺ T cells and CD4⁺ T cells are required for the control and resolution of productive MCMV infection in neonatal mice (Brizic et al. 2019). T cells are starting to control the virus around day 10 p.i. We have commented on these issues in the revised manuscript.

- The reduced production of NK cells in the bone marrow following perinatal infection is intriguing. Are NK cells the only affected lineage? It would be important to know whether other lineages are affected as well.

We have performed additional adoptive transfer experiments to answer this question and analyzed the numbers of T and B cells following adoptive transfer. In the revised manuscript, we show that the generation of B and T cells was not impaired. Therefore, the effect seems to be specific to NK cells. We have included this data in the revised manuscript (Fig. S6D).

- In the bone marrow transfer experiment (Figure 5J), was MCMV transferred too and were recipients mice infected following transfer? This information would help interpret the presented data.

To answer this question, we have isolated DNA from bone marrow cells isolated from mice on day 21 p.i. and performed qPCR to determine the presence of viral genomes. We have not detected MCMV genomes in the bone marrow, while at the same time, levels of MCMV DNA in the lungs of the same animals were high. Importantly, we analyzed if the virus replicates in the lungs and liver of bone marrow recipient mice to determine whether any virus is transferred and if it can reactivate. An infectious virus was not detected in these tissues indicating that the virus is not replicating after adoptive transfer. Of course, no method has absolute sensitivity, and it is possible that some very low levels of the virus were present in the tissues below the detection level of qPCR. However, based on our results, we do not believe any significant amount of virus was present. In addition, we routinely perform 40 cycles of qPCR to increase sensitivity. Our findings align with previously published data indicating that bone marrow cells from latently infected mice do not efficiently transmit the virus to recipients (Seckert et al. 2008). These data are now included in the revised manuscript (Fig. S6E-F).

Reviewer #2 (Remarks to the Author):

The study by Rožmanić et al reports interesting and novel findings regarding how viral infection early in life impacts NK cell development and antiviral immunity. The authors show that viral infection in young mice had a profound impact on NK cell function and maturation, associated with down-regulation of key transcription factor Eomes in terminally differentiated NK cells, with altered dysregulation of multiple transcription factors and cytokine expression. They provide evidence for the requirement of limited virus replication and, possibly, cytokines in this process. Major Comments:

This is a very interesting study and the concept that infection early in life detrimentally alters NK cell development is important. The majority of conclusions from the study are well-supported by the data. The aspect that is less well-developed is an understanding of the virus-induced factors that induce NK cell defects in vivo, and when:

In vitro-derived data demonstrating that the combination of IL-12 and IL-18 drive Eomes down-regulation is convincing. However, the authors only show IL-18 is up-regulated on day 21 post-infection in spleens upon MCMV infection. Is IL-12 also up-regulated? The kinetics of up-regulation of IL-18 at day 21 fits with KLRG1 induction at this time, but data in Figure 4, including experiments with therapeutic administration of antivirals, implies that early not late low-level virus replication is required for the NK cell dysfunction. Does early virus replication also lead to cytokine induction at these later time-points?

We agree that this is a very relevant point for the overall understanding of the study and that the terminally mature phenotype is the consequence of IL-12+IL-18 stimulation before day 21 p.i. We have also analyzed IL-12 expression, but in both mock and MCMV-infected mice, levels of IL-12 were below the detection limit. We have now stated this point in the revised manuscript. However, we have performed stimulation with IL-18 alone and now show that IL-18 alone is sufficient to drive KLRG1 upregulation on NK cells (Rebuttal figure 1A). To determine if therapeutic approaches affect IL-18 expression on day 21, we treated mice with ganciclovir and analyzed IL-18 gene expression in the spleen on day 21 (Rebuttal figure 1B-C). In line with the reviewer's comment, IL-18 levels in the ganciclovir-treated group were the same as in mock-infected mice, therefore arguing that earlier actions of inflammatory response drive NK cell terminal maturation. We have discussed these issues in the revised manuscript.

Rebuttal figure 1. (A) Splenic NK cells were isolated from naïve, 20 days old mice and were enriched using NK cell isolation kit (Miltenyi Biotec) and cultured in 50 ng/mL IL-15 for 5 days. Cells were then cultured in the presence of IL-18 (50 ng/mL) for 2 days. Quantification of KLRG1 expression by NK cells is shown. (B) Newborn C57BL/6 mice were infected with 200 PFU MCMV or mock-infected. The indicated group received i.p. ganciclovir every day post-infection. On days 14 and 21 p.i. expression of IL-18 was measured by qPCR in the bone marrow and spleen.

Why do the authors believe that Eomes is only down-regulated in mature NK cells during MCMV infection despite mature and immature NK cells expressing comparable Eomes levels (as per Fig. 3C)? Do they express altered expression of IL-12R/IL-18Rs as compared to immature cells? Does this alter between young and old mice? Or is there a more complex interaction related to difference between mature/immature NK cells in young versus old mice? Some explanation would improve the understanding of mechanism.

We thank the reviewer for pointing out that downregulation of Eomes is not directly related with NK cell maturation. We have rephrased these statements in the manuscript. *Differences in Eomes expression were primarily associated with the population of terminally differentiated NK cells (KLRG1+, CD27-CD11b+) (Fig. 4B and 4C). In contrast, less mature CD27+CD11b- NK cells in MCMV-infected mice downregulated Eomes expression in a minor subset of cells while the levels of Eomes in those that retained the expression were identical to that in NK cells from uninfected mice.* To determine IL-12R and IL-18R expression on mature and immature NK cells and compare expression of these receptors between adult and young mice, we have purchased monoclonal antibodies specific for these receptors and employed flow cytometry to address this issue experimentally. While the antibody specific for IL-18R performed well in our flow cytometry assays, the antibody specific for IL-12R did not. Therefore, we could only assess the levels of IL-18 receptor. We first determined the expression levels of IL-18R on NK cells in young, 10 and 21-day-old mice, and adult, 2-month-old uninfected mice. We show that the expression of IL-18R is significantly higher in NK cells in young mice (Fig. 6J and Fig. S6A). *Therefore, NK cells in young mice are more sensitive to IL-18 stimulation than NK cells in adult mice, potentially explaining the differential effect of IL-18 on NK cells in neonatal mice compared to adult mice.* Next, we compared IL-18R expression on NK cells in infected and control newborn mice and now show that MCMV infection causes IL-18R downregulation in NK cells. Furthermore, KLRG1+, and not KLRG1-, NK cells expressed low levels of IL-18R (Fig. S6B). The association of higher IL-18R expression in less mature human NK cells has been previously recognized (Goodier, Wolf and Riley 2020). Short-term activation of NK cells leads to IL-18R upregulation (Muller, Waldmann and Dubois 2014, Kunikata et al. 1998). Even stimulation of NK cells with IL-18 has been reported to induce IL-18R upregulation (Nielsen et al. 2016). However, IL-18 can also induce IL-18R downregulation (Hosohara et al. 2002). Thus, opposing effects of NK cell stimulation on IL-18R expression have been reported. We have included these new data in the revised manuscript, and have discussed it in the discussion section of the revised manuscript.

Fig 5C shows that Eomes expression by NK cells is required for accumulation of CD11b+CD27- NK cells. Paradoxically, Fig. 2C demonstrates increased CD11b+CD27- NK cell accumulation after MCMV infection and Fig 3 reports Eomes down-regulation upon MCMV infection. How do the authors explain these apparent conflicting data? Also, data presented in Fig 5C doesn't support the conclusion that 'these data demonstrate that conventional NK cells, not ILC1s, convert to terminally mature Eomeslow or Eomes- cells upon perinatal MCMV infection' and perhaps would be better suited elsewhere in the manuscript.

We apologize for not stating these issues clearly. We have included several sentences to explain these data in our initial manuscript. We have now expanded this part in the discussion section in the revised manuscript to make it clearer. Briefly, Eomes is essential for the development of NK cells (Zhang et al. 2018). Thus, mature NK cells cannot be generated without going through the step at which they express Eomes. However, once mature NK cells are formed, they can survive without Eomes, as shown by the use of the conditional knockout mice, i.e., tamoxifen-inducible deletion of Eomes (Madera et al. 2018). Based on our experiment using conditional knockout mice lacking Eomes

Rožmanić et al.
RESPONSE TO REVIEWERS

only in NKp46⁺ cells, which disrupts NK cell development, we are convinced that our claim that conventional NK cells convert to Eomes^{low} or Eomes⁻ NK cells is valid.

Minor Comments:

- Is UV virus not inflammatory? Does it induce cytokines in vivo? If not, presumably this is due to too few virions in preparations to induce measurable cytokine responses in vivo. This is worth showing/explaining.

UV-inactivated virus can trigger innate receptors (Doring et al. 2014) and stimulate immune response in mice (Vliegen et al. 2005). However, we agree with the reviewer that dosage could be determining factor. We have performed an experiment with 200 PFU, a dose we used throughout the manuscript, and a 1000-fold higher dose, 200 000 PFU, of UV-inactivated virus. In both cases, we got the same result, no impact on NK cells on post-natal day 21. We have included results for both doses in the revised manuscript (Fig. 5D). To assess the role of inflammatory response in general, we have injected PolyIC three times over the three-week period. Again, no impact on NK cells was observed (Fig. S5A). Thus, it seems that inflammatory stimuli alone cannot explain our findings.

- Fig 4E&F. These results, also with Fig. 4B&C, imply that early low-level replication is required for Eomes down-regulation. This is an important implication of the data and is worth stating. We agree with this remark and have thus introduced the importance of early replication in discussion section.

- Fig 1. Please show that NK1.1 deletion effectively led to deletion of NK cells to prove the point that this cell type is ineffective in early virus control. We have included flow cytometry data confirming that the depletion of NK cells in newborn mice was efficient (Fig. S1A).

- Line 188, please change 'is' to 'are'
We have corrected the text accordingly.

- Line 224, please introduce MCMV-MULT-1
We have added a description of MULT-1MCMV in the text.

- Presumably, the authors will provide the full datasets related to Fig 6?
The full datasets for Figure 1 and Figure 7 (ex Fig 6) will be made available, as stated in the revised manuscript.

- Statistics, multi-group analysis should be performed when there are more than two groups.
We have re-analyzed the data and included corrected statistics where needed in the revised manuscript to meet this suggestion.

Reviewer #3 (Remarks to the Author):

In this interesting paper by Rožmanić and colleagues, the impact of CMV on NK cells is examined. The particularly intriguing aspect of the paper is that it is examined in the context of a neonatal/perinatal infection model. This greatly adds to the value and interest of the paper. So many (nearly all) papers exploring the in vivo biology of NK/CMV interactions have done so in the context of a transplant/oncology paradigm, and yet the biggest impact of CMV on human health is congenital infection. This makes the paper particularly valuable, especially for pediatric physician-scientists. The model system the authors use is murine CMV (MCMV). Since MCMV does not cross the placenta, a neonatal infection model is used that mimics the pathologies of human congenital CMV. It is not perfect, but its relevance is convincing.

The authors found that MCMV infection of newborn mice severely impacted the NK function. The paper is valuable, the experiments well designed and well-controlled, and the conclusions supported by the data. The manuscript advances our understanding of the impact of congenital (presumably, recognizing the intrinsic challenges of a post-natal challenge model) CMV infection on NK cell biology.

A few suggestions are offered for ways the manuscript can be enhanced/clarified.

1. The infection had to be early in the neonatal period.

In C57BL/6 mice, the NK cell activating receptor Ly49H provides control of MCMV infection. NK cells do not express the Ly49H receptor in newborn mice, explaining the lack of NK cell dependent control of MCMV in the early period of life. Can the authors provide:

Rožmanić et al.
RESPONSE TO REVIEWERS

a. A clearer elucidation of when this deficit is overcome, i.e., what day of life?

To answer this question, we have infected 1-, 7- and 14-day-old mice with MCMV, and analyzed virus titer 4 days post-infection in control and NK cell-depleted groups of mice. This new data set shows that NK cells cannot control MCMV if mice are infected at 1 or 7 days after birth but can when mice are 14 days old. The ability of NK cells to control the virus correlates with the appearance of Ly49H⁺ NK cells. We have included these data in the revised manuscript (Fig. 1E and Fig. S1).

b. A clearer elucidation of how it is overcome, namely, the mechanism that transitions the neonatal mouse to a more "adult" immunophenotype?

NK cells are nearly absent in neonatal mice, as reported by Sparano et al. 2022, and therefore cannot control MCMV immediately. In the post-natal period, NK cells accumulate in the organs during ontogeny, which requires extensive NK cell proliferation (Jamieson et al. 2004). Furthermore, in the first weeks of life, NK cells do not express Ly49H, as we have now also shown in our model, which seems to be due to active suppression of NK cell maturation by TGF- β (Marcoe et al. 2012). *To expand these findings, we have performed transcriptomic analysis of sorted NK cells from young mice that are 7, 14 and 21 days old and from mice that are 60 days old (adult). A dynamic pattern of gene expression changes in NK cells over mouse life was observed (Fig. 1A). The major characteristic of NK cells in the post-natal period is highly increased expression of genes involved in the cell cycle, in line with the extensive process of proliferation required to fill organ niches during ontogeny. These findings were confirmed by analysis of Ki67 expression on post-natal days 14 and 60, which indicated drastically higher levels of Ki67 in NK cells of young mice (Fig. 1B). Furthermore, our transcriptomic data indicated that expression of genes coding for Ly49 receptors was low in NK cells of young mice, as compared to adult mice (Fig. 1C). Accordingly, expression of different Ly49 receptors, not just Ly49H, was absent on NK cells in the first several days of mouse life (Fig. 1D). Since Ly49 receptors mediate missing-self recognition and NK cell education/licensing (Rahim et al. 2014), our findings indicate that expression of Ly49 receptors marks the transition from neonatal to more adult NK cells which can control the infection. Importantly, these data are in accordance with observations that human fetal NK cells express lower levels of KIR receptors, functional homologs of Ly49 receptors (Ivarsson et al. 2013).*

c. Some perspective on other mouse strains. In other words, is this all unique to C57BL/6? What about BALB/c mice? This is not a trivial point with respect to the translational implications. If this is so highly strain specific that the mechanisms don't translate to different mouse strains, what is the "take-home" message for human health in the context of cCMV?

We agree with the reviewer that this is an important point when considering the relevance of our findings. We have included results for both the BALB/c and 129/SvJ mouse strains (Fig. 4D). Maturation and Eomes expression were affected in these strains similarly as in C57BL/6 mice, arguing against the major role of genetic background in the observed phenotype. Considering many similarities between our findings and observations made in humans, we are confident that our findings translate well into human health. Of course, further investigations are needed to expand on these findings, but that is beyond the scope of our manuscript.

2. The impact was evident even if replication-impaired viral mutants were used. The mutants described were MULT-1MCMV and Δ m74MCMV. The use of the UV inactivated virus that abrogates the effect completely is a nice control (Fig. 4D). One of these fails to make a gH/gL/gO complex. Other CMV systems suggest that gH is absolutely essential, but gO is dispensable. The authors should explain how any viral DNA replication can take place with a gH knockout. Isn't this incompatible with replication of any sort?

We have used gO-deficient MCMV (m74stop MCMV). For MCMV, two alternative gH/gL complexes, gH/gL/gO and gH/gL/MCK-2, have been identified (Lemmermann et al. 2015). While complex gH/gL/gO has a major role in establishing infection by efficient entry into different cells, this function can be achieved by the use of gH/gL/MCK-2 complex, even though to a much lower extent. Thus, gO-deficient MCMV can establish infection but achieves very low-level titers in different organs in mice (Lemmermann et al. 2015). We have elaborated on this issue in the revised manuscript.

3. Infection modified Eomes transcription, resulting in impaired NK cell function. The authors do a good job of elucidating the impact of Eomes on NK cells in this mouse system. The key point that they don't comment on in the discussion is human Eomes. Is the role of Eomes in humans similar to mice? Does perinatal/congenital infection have the same impact on NK cells in other mammalian systems, or with other pathogens (they do reference the Park paper, which might merit more comments - was the Park paper's toxoplasmosis work performed in C57/BL6 mice?)

Even though some differences in Eomes functions were reported recently (Wong et al. 2023), in general, the role of T-bet and Eomes seems to be very similar in human and mouse NK cells, as they

are in both cases required for development, differentiation and function of NK cells (Zhang et al. 2018). We have addressed this issue in the revised manuscript.

To assess if other infections can induce downregulation of Eomes, we infected newborn mice with the bacteria *Francisella tularensis* attenuated mutant Δ iglI ((Broms et al. 2011), Rebuttal figure 2). Bacterial infection induced terminal maturation of NK cells and Eomes downregulation to the same extent as MCMV infection. Thus, it seems this is a more general phenomenon; however, we feel that analysis of a greater number of pathogens is required before making any broader claims. Thus, we propose not to include this data in the manuscript. Importantly, our observations are in accordance with changes in NK cells upon congenital HCMV infection, as stated in the response to Reviewer 1.

Rebuttal figure 2. Downregulation of major transcription factors governing NK cell development is not MCMV specific. (A-C) Newborn C57Bl/6 mice were infected with 200 PFU of MCMV and 20 CFU of *Francisella IglI* mutant or mock-infected. On day 21, post-infection expression of CD27 and CD11b (A), KLRG1 and Eomes (B) was analyzed by flow cytometry. The frequency of terminally differentiated KLRG1 and Eomes expressing NK cells is shown (C). Mean values \pm SEM are shown (n=5-8; N= 1). Mann-Whitney test was used. *, $P < 0.05$; **, $P < 0.01$.

Yes, the Park study used C57Bl/6 mice. However, they were adults, not newborns (Park et al. 2019), and Eomes downregulation was observed. Our major claim is that in the case of MCMV infection, induced Eomes downregulation is specific for newborn mice and does not occur in adult mice. Furthermore, in the case of *Toxoplasma gondii* infection, NK cells convert to ILC1-like cells and express the prototypical marker of ILC1 cells, CD49a, which is not the case during MCMV infection of newborn mice. In addition, ILC1 cells induced by *Toxoplasma gondii* express high levels of activating receptors DNAM-1, which is severely diminished upon MCMV infection of newborn mice. On the other side, some similarities beyond reduced Eomes expression exist. In both cases, KLRG1 is upregulated, and Eomes can be regulated by IL-12. In the case of tumors, TGF- β has been shown to convert NK cells to ILC1-like cells distinct from those induced by *Toxoplasma gondii* infection (Gao et al. 2017). Collectively, these data illustrate the plasticity of NK cells in pathological conditions. We have commented on this issue in the discussion section of revised manuscript.

4. There was a major impact on a long non-coding RNA, IFNG-AS1, which appeared to mediate an effect on cytokine production. Figure 7D shows the qPCR data. Call me old-fashioned, but this reviewer would like to see this corroborated by Northern blot, to confirm that the altered PCR signal wasn't an artifact of changes in splicing or alternative promoter use of an upstream promoter. More to the point, what would be the mechanisms of this impact on transcription? The authors point out that it's a noncoding RNA. Is it even a pol II transcript? Is IFN gamma RNA the right control? How does IFNG-AS1 impact NK cell maturation? Is IFNG-AS1 conserved in humans? Again, the relevance to human cCMV biology is important to underscore here, since this mouse model is not truly a congenital infection model. Did, for example, the authors see any of the types of changes that are observed following early life and/or congenital CMV infections in infants (Vaaben AV, Levan J, Nguyen CBT, et al, doi:10.1093/infdis/jiac307).

The point about possible artifacts in qPCR is well taken. Indeed, our lab has previously employed Northern blots to resolve issues similar to the one raised by the reviewer. However, we would like to point out that our laboratories are equipped to handle only the non-radioactively labeled probes, and even when using the most sensitive materials and protocols available (Northern-Max Gly and

BrightStar membranes), the amount of RNA obtained from mouse NK cells is often insufficient for the Northern analysis. NK cells comprise only approximately 1% of total splenocytes in 21-day-old mice, of which a certain number is lost during fluorescence-activated cell sorting. We collected a maximum of 120000 cells from pooled spleens of 5 mice, and since lymphocytes are generally poor in RNA, we subsequently managed to isolate 27 ng of total RNA. Such an amount is well below the recommended for Northern analysis.

Nevertheless, as suggested by the reviewer, we attempted to detect *Ifng-as1* (and *Gapdh*) transcripts using Northern blot but could not detect any. We then considered increasing the number of mice for the collection of lymphocytes, but following 3R recommendations for minimizing the use of animals in scientific research, we would like to propose an alternative approach to respond to the issue raised by the reviewer. Namely, in addition to enabling differential expression analysis, our transcriptome data offers an unbiased overview of the transcription profile and exon usage in the *Ifng-as1* locus. As a reminder, the reads obtained after the sequencing of total polyA⁺ RNA isolated from KLRG1⁻ (mock), KLRG1⁻ (MCMV) and KLRG1⁺ (MCMV) NK cells, in addition to being used for transcript quantification, can also be aligned to the mouse genome and the location of mapped reads visualized using Integrative Genomics Viewer (IGV). We have used this approach to inspect the read coverage at the *Ifng-as1* locus in more detail. As can be seen from the IGV coverage track in the new Fig. S8, sequencing reads have almost exclusively been aligned within the exons (blue rectangles) of the *Ifng-as1* gene. This result is not surprising since the protocol used for the preparation of sequencing libraries selects for polyadenylated transcripts (mature transcripts that have already undergone splicing - a rapid co-transcriptional event that precedes polyadenylation); and it also suggests that *Ifng-as1* transcripts are indeed polyadenylated. More importantly, read coverage profiles of samples from KLRG1⁻ (mock), KLRG1⁻ (MCMV) and KLRG1⁺ (MCMV) NK cells are scaled to the same range on the y-axis in Supplementary Figure 8 (maximum of 120 reads per base). It is clearly visible that all exons of the *Ifng-as1* gene are equally downregulated in KLRG1⁺ NK cells obtained from MCMV-infected mice. Cutout framed with a red line shows a zoomed-in view of the 3' end of the *Ifng-as1* gene, in which the exon coverage in MCMV-infected KLRG1⁺ NK cells is more visible. We have also noted the location of *Ifng-as1* locus/transcripts examined by qPCR probes used in the manuscript, where one primer aligns to exon1 and another to exon2 while the probe spans the exon1-exon2 junction. In fact, the reviewer's comment is correct, as the qPCR assay appears insensitive to splice variant no. 2, which uses an alternative transcriptional start site. Nonetheless, as our sequencing data has shown, the conclusions we have drawn from the qPCR data as well as from quantification of transcript abundance that takes into consideration whole transcript length and splice variants - the amount of *Ifng-as1* lncRNA, all variants, is significantly reduced in KLRG1⁺ NK cells.

Regulation of IFN- γ expression is paramount for the host defense and prevention of immunopathology (Lees 2015). The mechanism of *Ifng-as1* as a regulator of IFNG locus is a subject of intensive research (Petermann et al. 2019). Both the *Ifng-as1* locus and *Ifng-as1* transcript modulate IFN- γ expression. The exact mechanism(s) of how the *Ifng-as1* transcript acts on IFN- γ expression is not known. However, it most likely acts on a post-transcriptional level by sequestering AU-binding proteins that destabilize RNA. In addition, we have performed additional experiments to investigate the mechanism driving NK cell *Ifng-as1* downregulation. *We show in the revised manuscript that prolonged IL-12+IL-18 stimulation, which we have identified as a factor governing NK cell maturation and Eomes downregulation, also drastically suppresses the expression of Ifng-as1. While this may seem counterintuitive, as IL-12+IL-18 are potent IFN- γ expression stimulators, this phenomenon depends on timing. Stein et al. have shown that IL-12+IL-18 drives strong upregulation of Ifng-as1 transcription in the first 6 hours of stimulation of NK cells (Stein et al. 2019). Similarly to mice, in human NK cells IFNG-AS1 regulates IFN- γ expression.*

As mentioned in response to reviewer 1, in utero congenital HCMV infection results in expansion of more mature NK cells without apparent differences in NK cell numbers compared to uninfected infants (Vaaben et al. 2022), in accordance with our study. In addition, NK cells of infected congenital cases have increased cytotoxic machinery and expand NKG2C⁺ NK cells, as we have observed in our model of perinatal infection, taking into account that there are parallels between NKG2C⁺ NK cells in humans and Ly49H⁺ NK cells in mice. Importantly, human NK cells of congenital cases downregulate CD7, indicative of IL-2 or IL-12+IL-18 stimulation. As shown in the revised manuscript, IL-12+IL-18 stimulation seems to be a major driver of NK cell transcriptional dysregulation, maturation and reduced functional capacity in our model. Altogether, this suggest that firm parallels exist between our model and congenital HCMV infection regarding its impact on NK cells.

5. There was an impact on the ability of the bone marrow to generate new NK cells. Figure 5 K-N is convincing on this point. The phenotype is transferable by adoptive transfer, which is interesting. Do the authors have a hypothesis as to why there is a reduction in bone marrow capacity to generate NK cells?

This question is of outstanding importance, and answering it will be a major direction in subsequent studies. *MCMV could affect bone marrow function either by infecting cells in the bone marrow, such*

as stromal cells, or by causing the production of inflammatory cytokines that can affect the function of bone marrow (Mori et al. 1999, Hirche et al. 2017). In the revised manuscript, we have included data showing that we do not detect viral genomes in the transferred bone marrow cells and no reactivation of the virus in recipient mice. Thus, transferred cells are unaffected by MCMV in recipient mice, implying that their state shaped in donor mice defines the generation of new cells. In addition, NK cells are not infected with MCMV. Therefore, we anticipate that inflammatory conditions in the bone marrow or consequences of stromal cell infection result in unfavorable conditions for immature and mature NK cell development. Our data suggest that the generation of immature NK cells is compromised, which can also result in reduced production of mature NK cells, which are drastically reduced in bone marrow. IL-15 is required for the homeostatic proliferation of NK cells (Castillo et al. 2009), and disturbances in the production of this cytokine or its signaling could impair the generation of NK cells. We have tested IL-15 and IL-15R α gene expression in the bone marrow and found slightly higher expression of both IL-15 and IL-15R α (Rebuttal figure 3), thus arguing against their involvement. In addition, we show in the revised manuscript that generation of B and T cells is not compromised significantly following adoptive transfer, implying that these effects are specific to NK cells. We have expanded the discussion to address this point.

Rebuttal Figure 3. Increased expression of IL-15 in the bone marrow of MCMV-infected newborn mice. (A-C) Newborn C57BL/6 mice were mock-infected or infected with 200 PFU of MCMV. On day 21 post-infection RNA was isolated from bone marrow, and RT-qPCR was performed to determine IL-15R α and IL-15 gene expression.

6. Antiviral treatment was unable to prevent these effects. This extends to immune globulin. Interestingly immune globulin has been shown to abrogate some of the developmental defects in these neonatal mice, including cerebellar hypoplasia, by these authors. The authors seemed to treat with immune sera and/or ganciclovir after viral challenge. The more interesting experiment would have been to treat BEFORE viral challenge. This would have obvious implications for CMV vaccines for congenital CMV infection.

Indeed, we have shown previously that antiviral antibodies can decrease pathological changes in the brain of infected neonatal mice (Cekinović et al. 2008). In our recent studies, we have demonstrated that infection-induced inflammatory response causes major pathological changes in the brain. We have also demonstrated that infection with attenuated recombinant MCMV, MULT-1MCMV, cannot induce an inflammatory response in the brain (Kveštak et al. 2021). However, unlike in the brain, infection with MULT-1MCMV causes deficits in NK cells (Figure 5C). These differences raise an intriguing possibility that NK cell deficits caused by infection, including very mild infection, can be present even in the absence of apparent clinical disease. We have expanded the discussion to address these issues.

We agree with the reviewer that understanding if vaccination approaches could prevent NK cell deficits caused by infection would be worth understanding. To answer this question, we performed maternal immunization by infecting female mice with MCMV and then infected their offspring (newborns) with MCMV. In this setup, newborn mice possess maternal antibodies which protect them against infection (Slavuljica et al. 2010). We have analyzed NK cell maturation and Eomes expression 21 days p.i. In the offspring of vaccinated mothers, NK cells were not terminally mature, and their Eomes was not downregulated. We have analyzed the virus in salivary glands to determine if the infection was established. Our results showed that the offspring of vaccinated female mice were protected against the establishment of infection, as no virus was detected in salivary glands. Altogether, these data argue that efficient vaccination could prevent the negative effects of viral infection in early life on NK cells. This data is included in the revised manuscript (Figure 5G).

References:

- Bennstein, S. B., S. Weinhold, A. R. Manser, N. Scherenschlich, A. Noll, K. Raba, G. Kogler, L. Walter & M. Uhrberg (2020) Umbilical cord blood-derived ILC1-like cells constitute a novel precursor for mature KIR(+)NKG2A(-) NK cells. *Elife*, 9.
- Brizic, I., L. Hirsl, M. Sustic, M. Golemac, W. J. Britt, A. Krmpotic & S. Jonjic (2019) CD4 T cells are required for maintenance of CD8 T(RM) cells and virus control in the brain of MCMV-infected newborn mice. *Med Microbiol Immunol*, 208, 487-494.
- Brizic, I., B. Lisnic, F. Krstanovic, W. Brune, H. Hengel & S. Jonjic (2022) Mouse Models for Cytomegalovirus Infections in Newborns and Adults. *Curr Protoc*, 2, e537.
- Broms, J. E., M. Lavander, L. Meyer & A. Sjostedt (2011) IgG and IgI of the Francisella pathogenicity island are important virulence determinants of Francisella tularensis LVS. *Infect Immun*, 79, 3683-96.
- Castillo, E. F., S. W. Stonier, L. Frasca & K. S. Schluns (2009) Dendritic cells support the in vivo development and maintenance of NK cells via IL-15 trans-presentation. *J Immunol*, 183, 4948-56.
- Cekinović, D., M. Golemac, E. P. Pugel, J. Tomac, L. Cicin-Sain, I. Slavuljica, R. Bradford, S. Misch, T. H. Winkler, M. Mach, W. J. Britt & S. Jonjić (2008) Passive immunization reduces murine cytomegalovirus-induced brain pathology in newborn mice. *J Virol*, 82, 12172-80.
- Collins, A., N. Rothman, K. Liu & S. L. Reiner (2017) Eomesodermin and T-bet mark developmentally distinct human natural killer cells. *JCI Insight*, 2, e90063.
- Doring, M., I. Lessin, T. Frenz, J. Spanier, A. Kessler, P. Tegtmeyer, F. Dag, N. Thiel, M. Trilling, S. Lienenklaus, S. Weiss, S. Scheu, M. Messerle, L. Cicin-Sain, H. Hengel & U. Kalinke (2014) M27 expressed by cytomegalovirus counteracts effective type I interferon induction of myeloid cells but not of plasmacytoid dendritic cells. *J Virol*, 88, 13638-50.
- Gao, Y., F. Souza-Fonseca-Guimaraes, T. Bald, S. S. Ng, A. Young, S. F. Ngiow, J. Rautela, J. Straube, N. Waddell, S. J. Blake, J. Yan, L. Bartholin, J. S. Lee, E. Vivier, K. Takeda, M. Messaoudene, L. Zitvogel, M. W. L. Teng, G. T. Belz, C. R. Engwerda, N. D. Huntington, K. Nakamura, M. Holzel & M. J. Smyth (2017) Tumor immunoevasion by the conversion of effector NK cells into type 1 innate lymphoid cells. *Nat Immunol*, 18, 1004-1015.
- Goodier, M. R., A. S. Wolf & E. M. Riley (2020) Differentiation and adaptation of natural killer cells for anti-malarial immunity. *Immunol Rev*, 293, 25-37.
- Hirche, C., T. Frenz, S. F. Haas, M. Döring, K. Borst, P. K. Tegtmeyer, I. Brizic, S. Jordan, K. Keyser, C. Chhatbar, E. Pronk, S. Lin, M. Messerle, S. Jonjic, C. S. Falk, A. Trumpp, M. A. G. Essers & U. Kalinke (2017) Systemic Virus Infections Differentially Modulate Cell Cycle State and Functionality of Long-Term Hematopoietic Stem Cells In Vivo. *Cell Rep*, 19, 2345-2356.
- Hosohara, K., H. Ueda, S. Kashiwamura, T. Yano, T. Ogura, S. Marukawa & H. Okamura (2002) Interleukin-18 induces acute biphasic reduction in the levels of circulating leukocytes in mice. *Clin Diagn Lab Immunol*, 9, 777-83.
- Ivarsson, M. A., L. Loh, N. Marquardt, E. Kekalainen, L. Berglin, N. K. Bjorkstrom, M. Westgren, D. F. Nixon & J. Michaelsson (2013) Differentiation and functional regulation of human fetal NK cells. *J Clin Invest*, 123, 3889-901.
- Jamieson, A. M., P. Isnard, J. R. Dorfman, M. C. Coles & D. H. Raulet (2004) Turnover and proliferation of NK cells in steady state and lymphopenic conditions. *J Immunol*, 172, 864-70.
- Koontz, T., M. Bralic, J. Tomac, E. Pernjak-Pugel, G. Bantug, S. Jonjic & W. J. Britt (2008) Altered development of the brain after focal herpesvirus infection of the central nervous system. *J Exp Med*, 205, 423-35.
- Kunikata, T., K. Torigoe, S. Ushio, T. Okura, C. Ushio, H. Yamauchi, M. Ikeda, H. Ikegami & M. Kurimoto (1998) Constitutive and induced IL-18 receptor expression by various peripheral blood cell subsets as determined by anti-hIL-18R monoclonal antibody. *Cell Immunol*, 189, 135-43.
- Kveštak, D., V. Juranić Lisnić, B. Lisnić, J. Tomac, M. Golemac, I. Brizić, D. Indenbirken, M. Cokarić Brdovčak, G. Bernardini, F. Krstanović, C. Rožmanić, A. Grundhoff, A. Krmpotić, W. J. Britt & S.

- Jonjić (2021) NK/ILC1 cells mediate neuroinflammation and brain pathology following congenital CMV infection. *J Exp Med*, 218.
- Lees, J. R. (2015) Interferon gamma in autoimmunity: A complicated player on a complex stage. *Cytokine*, 74, 18-26.
- Lemmermann, N. A., A. Krmpotic, J. Podlech, I. Brizic, A. Prager, H. Adler, A. Karbach, Y. Wu, S. Jonjic, M. J. Reddehase & B. Adler (2015) Non-redundant and redundant roles of cytomegalovirus gH/gL complexes in host organ entry and intra-tissue spread. *PLoS Pathog*, 11, e1004640.
- Madera, S., C. D. Geary, C. M. Lau, O. Pikovskaya, S. L. Reiner & J. C. Sun (2018) Cutting Edge: Divergent Requirement of T-Box Transcription Factors in Effector and Memory NK Cells. *J Immunol*, 200, 1977-1981.
- Marcoe, J. P., J. R. Lim, K. L. Schaubert, N. Fodil-Cornu, M. Matka, A. L. McCubbrey, A. R. Farr, S. M. Vidal & Y. Laouar (2012) TGF-beta is responsible for NK cell immaturity during ontogeny and increased susceptibility to infection during mouse infancy. *Nat Immunol*, 13, 843-50.
- Mori, T., M. Nakamura, K. Shimizu, Y. Ikeda & K. Ando (1999) In vivo disturbance of hematopoiesis in mice persistently infected with murine cytomegalovirus: impairment of stromal cell function. *Virology*, 253, 145-54.
- Moulden, J., C. Y. W. Sung, I. Brizic, S. Jonjic & W. Britt (2021) Murine Models of Central Nervous System Disease following Congenital Human Cytomegalovirus Infections. *Pathogens*, 10.
- Muller, J. R., T. A. Waldmann & S. Dubois (2014) Loss of cytotoxicity and gain of cytokine production in murine tumor-activated NK cells. *PLoS One*, 9, e102793.
- Nielsen, C. M., A. S. Wolf, M. R. Goodier & E. M. Riley (2016) Synergy between Common gamma Chain Family Cytokines and IL-18 Potentiates Innate and Adaptive Pathways of NK Cell Activation. *Front Immunol*, 7, 101.
- Noyola, D. E., A. Alarcon, A. Noguera-Julian, A. Muntasell, C. Munoz-Almagro, J. Garcia, A. Mur, C. Fortuny & M. Lopez-Botet (2015) Dynamics of the NK-cell subset redistribution induced by cytomegalovirus infection in preterm infants. *Hum Immunol*, 76, 118-23.
- Noyola, D. E., C. Fortuny, A. Muntasell, A. Noguera-Julian, C. Muñoz-Almagro, A. Alarcón, T. Juncosa, M. Moraru, C. Vilches & M. López-Botet (2012) Influence of congenital human cytomegalovirus infection and the NKG2C genotype on NK-cell subset distribution in children. *Eur J Immunol*, 42, 3256-66.
- Park, E., S. Patel, Q. Wang, P. Andhey, K. Zaitsev, S. Porter, M. Hershey, M. Bern, B. Plougastel-Douglas, P. Collins, M. Colonna, K. M. Murphy, E. Oltz, M. Artyomov, L. D. Sibley & W. M. Yokoyama (2019) Toxoplasma gondii infection drives conversion of NK cells into ILC1-like cells. *Elife*, 8.
- Petermann, F., A. Pekowska, C. A. Johnson, D. Jankovic, H. Y. Shih, K. Jiang, W. H. Hudson, S. R. Brooks, H. W. Sun, A. V. Villarino, C. Yao, K. Singleton, R. S. Akondy, Y. Kanno, A. Sher, R. Casellas, R. Ahmed & J. J. O'Shea (2019) The Magnitude of IFN-gamma Responses Is Fine-Tuned by DNA Architecture and the Non-coding Transcript of Ifng-as1. *Mol Cell*, 75, 1229-1242 e5.
- Rahim, M. M., M. M. Tu, A. B. Mahmoud, A. Wight, E. Abou-Samra, P. D. Lima & A. P. Makrigiannis (2014) Ly49 receptors: innate and adaptive immune paradigms. *Front Immunol*, 5, 145.
- Reddehase, M. J. & N. A. W. Lemmermann (2018) Mouse Model of Cytomegalovirus Disease and Immunotherapy in the Immunocompromised Host: Predictions for Medical Translation that Survived the "Test of Time". *Viruses*, 10.
- Seckert, C. K., A. Renzaho, M. J. Reddehase & N. K. Grzimek (2008) Hematopoietic stem cell transplantation with latently infected donors does not transmit virus to immunocompromised recipients in the murine model of cytomegalovirus infection. *Med Microbiol Immunol*, 197, 251-9.
- Slavuljica, I., A. Busche, M. Babić, M. Mitrović, I. Gašparović, D. Cekinović, E. Markova Car, E. Pernjak Pugel, A. Ciković, V. J. Lisnić, W. J. Britt, U. Koszinowski, M. Messerle, A. Krmpotic & S. Jonjić (2010) Recombinant mouse cytomegalovirus expressing a ligand for the NKG2D receptor is attenuated and has improved vaccine properties. *J Clin Invest*, 120, 4532-45.

Rožmanić et al.
RESPONSE TO REVIEWERS

- Stein, N., O. Berhani, D. Schmiedel, A. Duev-Cohen, E. Seidel, I. Kol, P. Tsukerman, M. Hecht, A. Reches, M. Gamliel, A. Obeidat, Y. Charpak-Amikam, R. Yamin & O. Mandelboim (2019) IFNG-AS1 Enhances Interferon Gamma Production in Human Natural Killer Cells. *iScience*, 11, 466-473.
- Sun, J. C., J. N. Beilke & L. L. Lanier (2009) Adaptive immune features of natural killer cells. *Nature*, 457, 557-61.
- Uksila, J., O. Lassila, T. Hirvonen & P. Toivanen (1983) Development of natural killer cell function in the human fetus. *J Immunol*, 130, 153-6.
- Vaaben, A. V., J. Levan, C. B. T. Nguyen, P. C. Callaway, M. Prah, L. Warriar, F. Nankya, K. Musinguzi, A. Kakuru, M. K. Muhindo, G. Dorsey, M. R. Kamya & M. E. Feeney (2022) In Utero Activation of Natural Killer Cells in Congenital Cytomegalovirus Infection. *J Infect Dis*, 226, 566-575.
- Vliegenhart, I., S. B. Herengreen, G. E. Grauls, C. A. Bruggeman & F. R. Stassen (2005) Mouse cytomegalovirus antigenic immune stimulation is sufficient to aggravate atherosclerosis in hypercholesterolemic mice. *Atherosclerosis*, 181, 39-44.
- Wong, P., J. A. Foltz, L. Chang, C. C. Neal, T. Yao, C. C. Cubitt, J. Tran, S. Kersting-Schadek, S. Palakurty, N. Jaeger, D. A. Russler-Germain, N. D. Marin, M. Gang, J. A. Wagner, A. Y. Zhou, M. T. Jacobs, M. Foster, T. Schappe, L. Marsala, E. McClain, P. Pence, M. Becker-Hapak, B. Fisk, A. A. Petti, O. L. Griffith, M. Griffith, M. M. Berrien-Elliott & T. A. Fehniger (2023) T-BET and EOMES sustain mature human NK cell identity and antitumor function. *J Clin Invest*, 133.
- Zhang, J., M. Marotel, S. Fauteux-Daniel, A. L. Mathieu, S. Viel, A. Marçais & T. Walzer (2018) T-bet and Eomes govern differentiation and function of mouse and human NK cells and ILC1. *Eur J Immunol*, 48, 738-750.

Reviewers' Comments:

Reviewer #1:

Remarks to the Author:

The authors have extensively reviewed the manuscript and have adequately addressed the queries and comments of the reviewer. Additional data have been included that further enhance the value of the manuscript. The discussion section is more balanced and nicely puts the results into perspective.

Reviewer #2:

Remarks to the Author:

The authors have made extensive efforts to address my comments in the revision, and the paper is much improved and now provided substantially increased mechanistic insight. This is a high-quality, interesting and important paper.

Reviewer #3:

Remarks to the Author:

Major Points

In this revised manuscript, Rožmanić et al. have been highly responsive to the previous critiques of the reviewers.

I am particularly impressed by the fact that the authors have performed additional experiments, as in Fig. 2A, Fig. S2, Fig. 1E, Fig. S1, and heroic adoptive transfer studies (Fig. S6D) to address points raised by the reviewers.

Reviewer 1 brought up issues about the differences intrinsic between human and NK cell ontogeny. These are important points. Such are the intrinsic limitations in animal models of infection. I believe that Rožmanić has been highly responsive to this particular criticism and, at the end of the day, the key question is, do the authors acknowledge the limitations of their model and caution potential readers of the manuscript not to too broadly draw conclusions about the direct translational relevance of their findings to human health? I believe the answer to both questions is "yes". I would agree with the authors' statement that "...firm parallels exist between our model and congenital HCMV infection" regarding its impact on NK cells". Appropriate limitations are acknowledged.

Reviewer 2 brought up the issue of a need for an increased understanding of the virus-induced factors that induce NK cell defects in vivo. To address this, the authors clarify the original observation that the combination of IL-12 and IL-18 drive Eomes downregulation, by providing more data making it clear that it is, in fact, IL-18 that is the major driver of this phenotype. The additional data is clearly presented, and convincing (rebuttal figure 1A). The experiment with ganciclovir further supports the authors' assertion that it is an early (or possibly an immediate early?) effect (Rebuttal figure 1B-C).

My only minor criticism here is that the ganciclovir experiment uses an immunophenotype as the "read-out", and the authors infer that the effect must be mediated at an early stage of the viral life-cycle. But, without knowing the key MCMV gene product(s) involved (which would permit quantitative reverse-transcriptase PCR with assignment of the putative regulatory transcript(s) to attendant kinetic class), it's impossible to know with certainty how ganciclovir is impacting this at the molecular and transcriptional level. I don't think the authors need to do additional experiments at this point - I am sure this would be an interesting line of study in its own right, to identify the regulatory viral gene, make a mutant, map the mechanism - but the authors should at least acknowledge the point and identify this as an area for future study. Or, if the authors have a hypothesis as to the key regulatory viral gene, they should state what it is.

Reviewer 3 points out how the neonatal mouse is the ontological equivalent of the human fetus in many ways and asks for clarification as to the critical window(s) in development that are elucidated. In response, the authors performed more experiments (Fig. 1E and Fig. S1) that show NK cells cannot control MCMV if mice are infected at 1 or 7 days after birth but can when mice are 14 days old. This is interesting. Although such a "conversion scale" may not exist, it would be interesting to know how these post-natal dates correspond to the age of a human fetus. Reviewer 3 also pointed out the mouse strain-to-strain variation that's been seen over many years of research in response to MCMV, and the authors now provide reassuring data from BALB/c and 129/SvJ mouse strains (Fig. 4D). Finally, this reviewer brought up the toxoplasmosis data as it relates to Eomes and NK cells and, to their credit, infections can induce downregulation of Eomes, the investigators infected newborn mice with *Francisella tularensis* attenuated mutant Δ iglI and showed that this induced terminal maturation of NK cells and Eomes downregulation to the same extent as MCMV infection. Thus, it indeed seems this is a more general phenomenon. The authors also do a good job responding to the comments about transcriptional analysis, and the role of maternal immunity in abrogating the impact on NK cells in the neonate. The performed maternal immunization by infecting female mice with MCMV and then infecting newborn mice with MCMV. NK cell maturation and Eomes expression 21 days p.i. in the offspring of vaccinated mothers showed that NK cells were not terminally mature, and their Eomes expression was not downregulated. Thus, vaccination prevents the negative effects of viral infection in early life on NK cells (Figure 5G). These new experiments are just outstanding, a brilliant response to the reviewers, and is the kind of thing that makes this worthy of a Nature Communications paper. The authors have done an outstanding job in response to the reviews and are to be congratulated, in this reviewer's opinion, for this terrific work!

Rožmanić et al.
RESPONSE TO REVIEWERS

Dear Dr. Mason and dear reviewers,

Thank you for the positive comments, time invested, and for accepting (in principle) our manuscript for publication. In the revised manuscript, we have addressed remaining concerns. Below we provide point-by-point response to reviewers (in blue), and we hope that our manuscript is now suitable for publication in Nature Communications.

Reviewer #1 (Remarks to the Author):

The authors have extensively reviewed the manuscript and have adequately addressed the queries and comments of the reviewer. Additional data have been included that further enhance the value of the manuscript. The discussion section is more balanced and nicely puts the results into perspective.

-

Reviewer #2 (Remarks to the Author):

The authors have made extensive efforts to address my comments in the revision, and the paper is much improved and now provided substantially increased mechanistic insight. This is a high-quality, interesting and important paper.

-

Reviewer #3 (Remarks to the Author):

Major Points

In this revised manuscript, Rožmanić et al. have been highly responsive to the previous critiques of the reviewers.

I am particularly impressed by the fact that the authors have performed additional experiments, as in Fig. 2A, Fig. S2, Fig. 1E, Fig. S1, and heroic adoptive transfer studies (Fig. S6D) to address points raised by the reviewers.

Reviewer 1 brought up issues about the differences intrinsic between human and NK cell ontogeny. These are important points. Such are the intrinsic limitations in animal models of infection. I believe that Rožmanić has been highly responsive to this particular criticism and, at the end of the day, the key question is, do the authors acknowledge the limitations of their model and caution potential readers of the manuscript not to too broadly draw conclusions about the direct translational relevance of their findings to human health? I believe the answer to both questions is "yes". I would agree with the authors' statement that "...firm parallels exist between our model and congenital HCMV infection" regarding its impact on NK cells". Appropriate limitations are acknowledged.

Reviewer 2 brought up the issue of a need for an increased understanding of the virus-induced factors that induce NK cell defects in vivo. To address this, the authors clarify the original observation that the combination of IL-12 and IL-18 drive Eomes downregulation, by providing more data making it clear that it is, in fact, IL-18 that is the major driver of this phenotype. The additional data is clearly presented, and convincing (rebuttal figure 1A). The experiment with ganciclovir further supports the authors' assertion that it is an early (or possibly an immediate early?) effect (Rebuttal figure 1B-C). My only minor criticism here is that the ganciclovir experiment uses an immunophenotype as the "read-out", and the authors infer that the effect must be mediated at an early stage of the viral life-cycle. But, without knowing the key MCMV gene product(s) involved (which would permit quantitative reverse-transcriptase PCR with assignment of the putative regulatory transcript(s) to attendant kinetic class), it's impossible to know with certainty how ganciclovir is impacting this at the molecular and transcriptional level. I don't think the authors need to do additional experiments at this point - I am sure this would be an interesting line of study in its own right, to identify the regulatory viral gene, make a mutant, map the mechanism - but the authors should at least acknowledge the point and identify this as an area for future study. Or, if the authors have a hypothesis as to the key regulatory viral gene, they should state what it is.

We have used ganciclovir to demonstrate that currently available treatments which reduce the level of virus cannot affect the impact of infection on NK cells. Since we can recover infectious virus from different tissues of ganciclovir treated mice, this implies that some virus replication is ongoing despite ganciclovir treatment, i.e. that some virus will go through the whole lifecycle. Whether any particular MCMV gene(s) contribute to observed phenotype of NK cells remains to be determined in the future studies. We have made small modifications in the main text to address this issue.

Rožmanić et al.
RESPONSE TO REVIEWERS

Reviewer 3 points out how the neonatal mouse is the ontological equivalent of the human fetus in many ways and asks for clarification as to the critical window(s) in development that are elucidated. In response, the authors performed more experiments (Fig. 1E and Fig. S1) that show NK cells cannot control MCMV if mice are infected at 1 or 7 days after birth but can when mice are 14 days old. This is interesting. Although such a "conversion scale" may not exist, it would be interesting to know how these post-natal dates correspond to the age of a human fetus.

To clarify this issue, we have included additional information in the discussion section, to state that the 7-10 days old mice correspond to human newborn immunologically (Mold and McCune 2012). However, conversion scale for specific immune cells is at the moment unclear. For example, while it is published that certain aspects of T cell development in newborn mice correspond to second trimester fetus in humans, NK cells appear during late gestation/postnatally in mice, and first trimester in humans. We have introduced these facts into manuscript.

Reviewer 3 also pointed out the mouse strain-to-strain variation that's been seen over many years of research in response to MCMV, and the authors now provide reassuring data from BALB/c and 129/SvJ mouse strains (Fig. 4D). Finally, this reviewer brought up the toxoplasmosis data as it relates to Eomes and NK cells and, to their credit, infections can induce downregulation of Eomes, the investigators infected newborn mice with Francisella tularensis attenuated mutant Δ iglI and showed that this induced terminal maturation of NK cells and Eomes downregulation to the same extent as MCMV infection. Thus, it indeed seems this is a more general phenomenon. The authors also do a good job responding to the comments about transcriptional analysis, and the role of maternal immunity in abrogating the impact on NK cells in the neonate. The performed maternal immunization by infecting female mice with MCMV and then infecting newborn mice with MCMV. NK cell maturation and Eomes expression 21 days p.i. in the offspring of vaccinated mothers showed that NK cells were not terminally mature, and their Eomes expression was not downregulated. Thus, vaccination prevents the negative effects of viral infection in early life on NK cells (Figure 5G). These new experiments are just outstanding, a brilliant response to the reviewers, and is the kind of thing that makes this worthy of a Nature Communications paper. The authors have done an outstanding job in response to the reviews and are to be congratulated, in this reviewer's opinion, for this terrific work!

We thank all the reviewers again for their comments and support. We feel that due to our mutual efforts the manuscript is now significantly improved.

References:

Mold, J. E. & J. M. McCune (2012) Immunological tolerance during fetal development: from mouse to man. *Adv Immunol*, 115, 73-111.